# Towards Understanding Grokking:
# An Effective Theory of Representation Learning

**Ziming Liu, Ouail Kitouni, Niklas Nolte, Eric J. Michaud, Max Tegmark, Mike Williams**
Department of Physics, Institute for AI and Fundamental Interactions, MIT
`{zmliu,kitouni,nnolte,ericjm,tegmark,mwill}@mit.edu`

## Abstract

We aim to understand *grokking*, a phenomenon where models generalize long after overfitting their training set. We present both a *microscopic* analysis anchored by an effective theory and a *macroscopic* analysis of phase diagrams describing learning performance across hyperparameters. We find that generalization originates from structured representations whose training dynamics and dependence on training set size can be predicted by our effective theory in a toy setting. We observe empirically the presence of four learning phases: *comprehension*, *grokking*, *memorization*, and *confusion*. We find representation learning to occur only in a "Goldilocks zone" (including comprehension and grokking) between memorization and confusion. We find on transformers the grokking phase stays closer to the memorization phase (compared to the comprehension phase), leading to delayed generalization. The Goldilocks phase is reminiscent of "intelligence from starvation" in Darwinian evolution, where resource limitations drive discovery of more efficient solutions. This study not only provides intuitive explanations of the origin of grokking, but also highlights the usefulness of physics-inspired tools, e.g., effective theories and phase diagrams, for understanding deep learning.

## 1 Introduction

Perhaps *the* central challenge of a scientific understanding of deep learning lies in accounting for neural network generalization. Power et al. [1] recently added a new puzzle to the task of understanding generalization with their discovery of *grokking*. Grokking refers to the surprising phenomenon of *delayed generalization* where neural networks, on certain learning problems, generalize long after overfitting their training set. It is a rare albeit striking phenomenon that violates common machine learning intuitions, raising three key puzzles:

**Q1** *The origin of generalization*: When trained on the algorithmic datasets where grokking occurs, how do models generalize at all?

**Q2** *The critical training size*: Why does the training time needed to "grok" (generalize) diverge as the training set size decreases toward a critical point?

**Q3** *Delayed generalization*: Under what conditions does delayed generalization occur?

We provide evidence that representation learning is central to answering each of these questions. Our answers can be summarized as follows:

**A1** Generalization can be attributed to learning a good representation of the input embeddings, i.e., a representation that has the appropriate structure for the task and which can be predicted from the theory in Section 3. See Figures 1 and 2.

**A2** The critical training set size corresponds to the least amount of training data that can determine such a representation (which, in some cases, is unique up to linear transformations).

36th Conference on Neural Information Processing Systems (NeurIPS 2022).

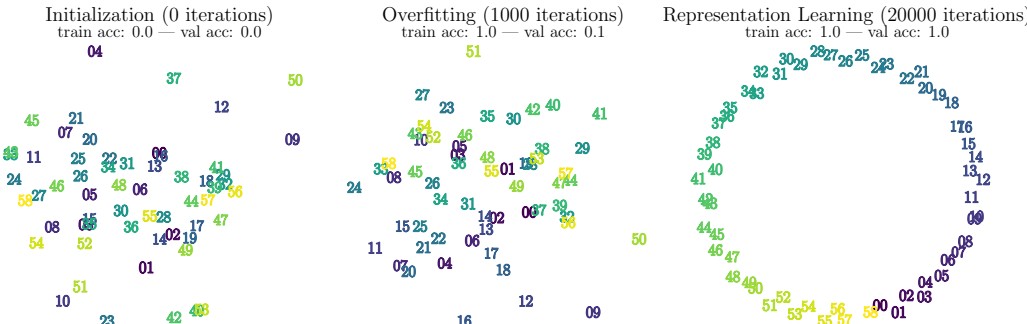

Figure 1: Visualization of the first two principal components of the learned input embeddings at different training stages of a transformer learning modular addition. We observe that generalization coincides with the emergence of structure in the embeddings. See Section 4.2 for the training details.

**A3** Grokking is a phase between "comprehension" and "memorization" phases and it can be remedied with proper hyparmeter tuning, as illustrated by the phase diagrams in Figure 6.

This paper is organized as follows: In Section 2, we introduce the problem setting and build a simplified toy model. In Section 3, we will use an *effective theory* approach, a useful tool from theoretical physics, to shed some light on questions **Q1** and **Q2** and show the relationship between generalization and the learning of structured representations. In Section 4, we explain **Q3** by displaying phase diagrams from a grid search of hyperparameters and show how we can "de-delay" generalization by following intuition developed from the phase diagram. We discuss related work in Section 5, followed by conclusions in Section 6.[1]

## 2 Problem Setting

Power et al. [1] observe grokking on a less common task – learning "algorithmic" binary operations. Given some binary operation $\circ$, a network is tasked with learning the map $(a, b) \mapsto c$ where $c = a \circ b$. They use a decoder-only transformer to predict the second to last token in a tokenized equation of the form "<lhs> <op> <rhs> <eq> **<result>** <eos>". Each token is represented as a 256-dimensional embedding vector. The embeddings are learnable and initialized randomly. After the transformer, a final linear layer maps the output to class logits for each token.

**Toy Model** We primarily study grokking in a simpler toy model, which still retains the key behaviors from the setup of [1]. Although [1] treated this as a classification task, we study both regression (mean-squared error) and classification (cross-entropy). The basic setup is as follows: our model takes as input the symbols $a, b$ and maps them to trainable embedding vectors $\mathbf{E}_a, \mathbf{E}_b \in \mathbb{R}^{d_{\text{in}}}$. It then sums $\mathbf{E}_a, \mathbf{E}_b$ and sends the resulting vector through a "decoder" MLP. The target output vector, denoted $\mathbf{Y}_c \in \mathbb{R}^{d_{\text{out}}}$ is a fixed random vector (regression task) or a one-hot vector (classification task). Our model architecture can therefore be compactly described as $(a, b) \mapsto \text{Dec}(\mathbf{E}_a + \mathbf{E}_b)$, where the embeddings $\mathbf{E}_*$ and the decoder are trainable. Despite its simplicity, this toy model can generalize to all abelian groups (discussed in Appendix B). In sections 3-4.1, we consider only the binary operation of addition. We consider modular addition in Section 4.2 to generalize some of our results to a transformer architecture and study general non-abelian operations in Appendix H.

**Dataset** In our toy setting, we are concerned with learning the addition operation. A data sample corresponding to $i + j$ is denoted as $(i, j)$ for simplicity. If $i, j \in \{0, \dots, p - 1\}$, there are in total $p(p + 1)/2$ different samples since we consider $i + j$ and $j + i$ to be the same sample. A dataset $D$ is a set of non-repeating data samples. We denote the full dataset as $D_0$ and split it into a training dataset $D$ and a validation dataset $D'$, i.e., $D \bigcup D' = D_0, D \bigcap D' = \emptyset$. We define *training data fraction* $= |D|/|D_0|$ where $|\cdot|$ denotes the cardinality of the set.

---

[1] Project code can be found at: https://github.com/ejmichaud/grokking-squared

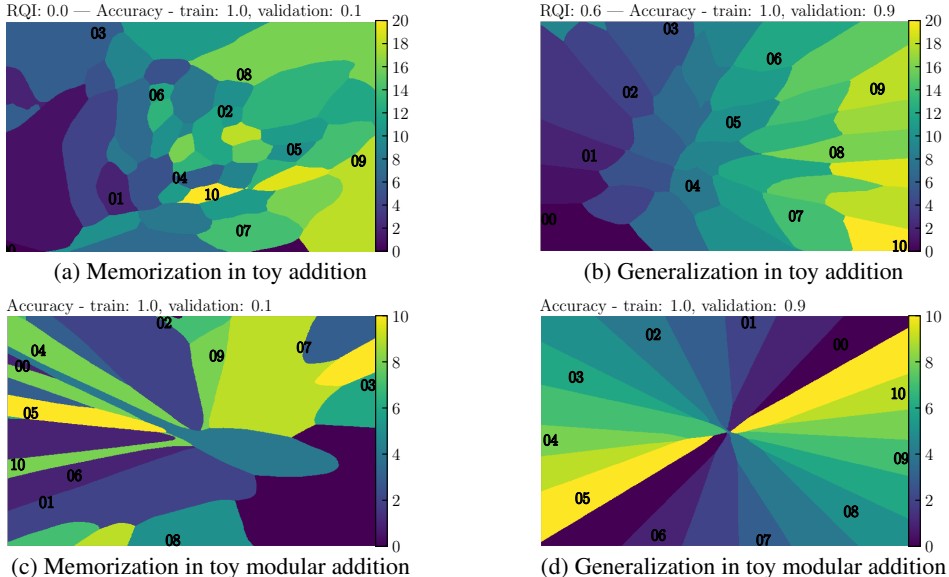

Figure 2: Visualization of the learned set of embeddings ($p = 11$) and the decoder function associated with it for the case of 2D embeddings. Axes refer to each dimension of the learned embeddings. The decoder is evaluated on a grid of points in embedding-space and the color at each point represents the highest probability class. For visualization purposes, the decoder is trained on inputs of the form $(\mathbf{E}_i + \mathbf{E}_j)/2$. One can read off the output of the decoder when fed the operation $i \circ j$ from this figure simply by taking the midpoint between the respective embeddings of $i$ and $j$.

## 3 Why Generalization Occurs: Representations and Dynamics

We can see that generalization appears to be linked to the emergence of highly-structured embeddings in Figure 2. In particular, Figure 2 (a, b) shows parallelograms in toy addition, and (c, d) shows a circle in toy modular addition. We now restrict ourselves to the toy addition setup and formalize a notion of representation quality and show that it predicts the model's performance. We then develop a physics-inspired *effective* theory of learning which can accurately predict the critical training set size and training trajectories of representations. The concept of an effective theory in physics is similar to model reduction in computational methods in that it aims to describe complex phenomena with simple yet intuitive pictures. In our effective theory, we will model the dynamics of representation learning not as gradient descent of the true task loss but rather a simpler effective loss function $\ell_{\text{eff}}$ which depends only on the representations in embedding space and not on the decoder.

### 3.1 Representation quality predicts generalization for the toy model

A rigorous definition for *structure* in the learned representation is necessary. We propose the following definition,

**Definition 1.** $(i, j, m, n)$ *is a $\delta$-**parallelogram** in the representation* $\mathbf{R} \equiv [\mathbf{E}_0, \cdots, \mathbf{E}_{p-1}]$ *if*

$$|(\mathbf{E}_i + \mathbf{E}_j) - (\mathbf{E}_m + \mathbf{E}_n)| \leq \delta.$$

In the following derivations, we can take $\delta$, which is a small threshold to tolerate numerical errors, to be zero.

**Proposition 1.** *When the training loss is zero, any parallelogram $(i, j, m, n)$ in representation $\mathbf{R}$ satisfies $i + j = m + n$.*

*Proof.* Suppose that this is not the case, i.e., suppose $\mathbf{E}_i + \mathbf{E}_j = \mathbf{E}_m + \mathbf{E}_n$ but $i + j \neq m + n$, then $\mathbf{Y}_{i+j} = \text{Dec}(\mathbf{E}_i + \mathbf{E}_j) = \text{Dec}(\mathbf{E}_m + \mathbf{E}_n) = \mathbf{Y}_{m+n}$ where the first and last equalities come from the zero training loss assumption. However, since $i + j \neq m + n$, we have $\mathbf{Y}_{i+j} \neq \mathbf{Y}_{n+m}$ (almost surely in the regression task), a contradiction. $\square$

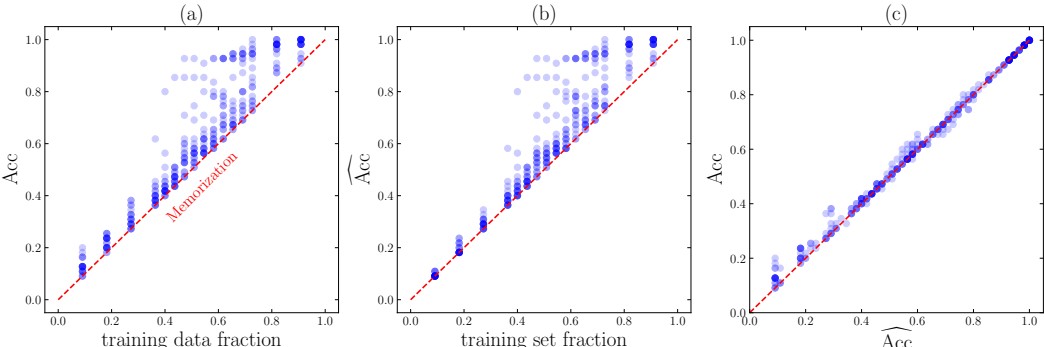

Figure 3: We compute accuracy (of the full dataset) either measured empirically Acc, or predicted from the representation of the embeddings $\widehat{\mathrm{Acc}}$. These two accuracies as a function of training data fraction are plotted in (a)(b), and their agreement is shown in (c).

It is convenient to define the permissible parallelogram set associated with a training dataset $D$ ("permissible" means consistent with 100% training accuracy) as

$$P_0(D) = \{(i,j,m,n)|(i,j) \in D, (m,n) \in D, i+j = m+n\}. \tag{1}$$

For simplicity, we denote $P_0 \equiv P_0(D_0)$. Given a representation $\mathbf{R}$, we can check how many permissible parallelograms actually exist in $\mathbf{R}$ within error $\delta$, so we define the parallelogram set corresponding to $\mathbf{R}$ as

$$P(\mathbf{R}, \delta) = \{(i,j,m,n)|(i,j,m,n) \in P_0, |(\mathbf{E}_i + \mathbf{E}_j) - (\mathbf{E}_m + \mathbf{E}_n)| \le \delta\}. \tag{2}$$

For brevity we will write $P(\mathbf{R})$, suppressing the dependence on $\delta$. We define the representation quality index (RQI) as

$$\mathrm{RQI}(\mathbf{R}) = \frac{|P(\mathbf{R})|}{|P_0|} \in [0, 1]. \tag{3}$$

We will use the term *linear representation* or *linear structure* to refer to a representation whose embeddings are of the form $\mathbf{E}_k = \mathbf{a} + k\mathbf{b}$ ($k = 0, \cdots, p-1; \mathbf{a}, \mathbf{b} \in \mathbb{R}^{d_{\mathrm{in}}}$). A linear representation has $\mathrm{RQI} = 1$, while a random representation (sampled from, say, a normal dstribution) has $\mathrm{RQI} = 0$ with high probability.

Quantitatively, we denote the "predicted accuracy" $\widehat{\mathrm{Acc}}$ as the accuracy achievable on the whole dataset given the representation $\mathbf{R}$ (see Appendix D for the full details). In Figure 3, we see that the predicted $\widehat{\mathrm{Acc}}$ aligns well with the true accuracy Acc, establishing good evidence that structured representation of input embeddings leads to generalization. We use an example to illustrate the origin of generalization here. In the setup of Figure 2 (b), suppose the decoder can achieve zero training loss and $\mathbf{E}_6 + \mathbf{E}_8$ is a training sample hence $\mathrm{Dec}(\mathbf{E}_6 + \mathbf{E}_8) = \mathbf{Y}_{14}$. At validation time, the decoder is tasked with predicting a validation sample $\mathbf{E}_5 + \mathbf{E}_9$. Since $(5, 9, 6, 8)$ forms a parallelogram such that $\mathbf{E}_5 + \mathbf{E}_9 = \mathbf{E}_6 + \mathbf{E}_8$, the decoder can predict the validation sample correctly because $\mathrm{Dec}(\mathbf{E}_5 + \mathbf{E}_9) = \mathrm{Dec}(\mathbf{E}_6 + \mathbf{E}_8) = \mathbf{Y}_{14}$.

### 3.2 The dynamics of embedding vectors

Suppose that we have an ideal model $\mathcal{M}^* = (\mathrm{Dec}^*, \mathbf{R}^*)$ such that:[2]

- (1) $\mathcal{M}^*$ can achieve zero training loss;
- (2) $\mathcal{M}^*$ has an injective decoder, i.e., $\mathrm{Dec}^*(\mathbf{x}_1) \ne \mathrm{Dec}^*(\mathbf{x}_2)$ for any $\mathbf{x}_1 \ne \mathbf{x}_2$.

Then Proposition 2 provides a mechanism for the formation of parallelograms.

---

[2]One can verify a posteriori if a trained model $\mathcal{M}$ is close to being an ideal model $\mathcal{M}^*$. Please refer to Appendix E for details.

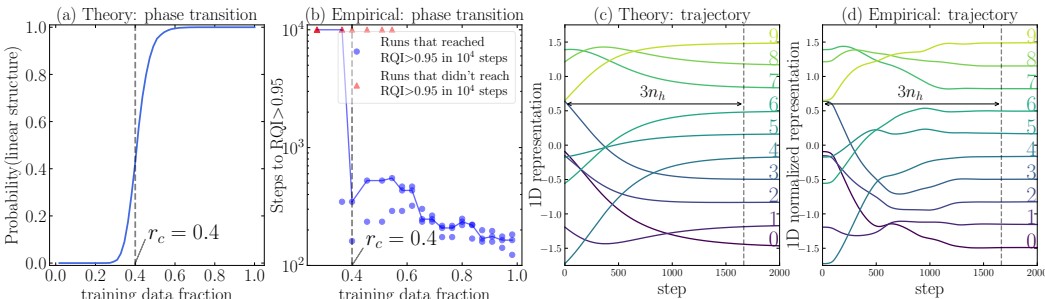

Figure 4: (a) The effective theory predicts a phase transition in the probability of obtaining a linear representation around $r_c = 0.4$. (b) Empirical results display a phase transition of RQI around $r_c = 0.4$, in agreement with the theory (the blue line shows the median of multiple random seeds). The evolution of 1D representations predicted by the effective theory or obtained from neural network training (shown in (c) and (d) respectively) agree creditably well.

**Proposition 2.** *If a training set $D$ contains two samples $(i,j)$ and $(m,n)$ with $i + j = m + n$, then $\mathcal{M}^*$ learns a representation $\mathbf{R}^*$ such that $\mathbf{E}_i + \mathbf{E}_j = \mathbf{E}_m + \mathbf{E}_n$, i.e., $(i,j,m,n)$ forms a parallelogram.*

*Proof.* Due to the zero training loss assumption, we have $\mathrm{Dec}^*(\mathbf{E}_i + \mathbf{E}_j) = \mathbf{Y}_{i+j} = \mathbf{Y}_{m+n} = \mathrm{Dec}^*(\mathbf{E}_m + \mathbf{E}_n)$. Then the injectivity of $\mathrm{Dec}^*$ implies $\mathbf{E}_i + \mathbf{E}_j = \mathbf{E}_m + \mathbf{E}_n$. $\qquad\square$

The dynamics of the trained embedding vectors are determined by various factors interacting in complex ways, for instance: the details of the decoder architecture, the optimizer hyperparameters, and the various kinds of implicit regularization induced by the training procedure. We will see that the dynamics of normalized quantities, namely, the normalized embeddings at time $t$, defined as $\tilde{\mathbf{E}}_k^{(t)} = \frac{\mathbf{E}_k^{(t)} - \mu_t}{\sigma_t}$, where $\mu_t = \frac{1}{p}\sum_k \mathbf{E}_k^{(t)}$ and $\sigma_t = \frac{1}{p}\sum_k |\mathbf{E}_k^{(t)} - \mu_t|^2$, can be qualitatively described by a simple effective loss (in the physics effective theory sense). We will assume that the normalized embedding vectors obey a gradient flow for an effective loss function of the form

$$\frac{d\tilde{\mathbf{E}}_i}{dt} = -\frac{\partial \ell_{\mathrm{eff}}}{\partial \tilde{\mathbf{E}}_i}, \tag{4}$$

$$\ell_{\mathrm{eff}} = \frac{\ell_0}{Z_0}, \quad \ell_0 \equiv \sum_{(i,j,m,n)\in P_0(D)} |\tilde{\mathbf{E}}_i + \tilde{\mathbf{E}}_j - \tilde{\mathbf{E}}_m - \tilde{\mathbf{E}}_n|^2 / |P_0(D)|, \quad Z_0 \equiv \sum_k |\tilde{\mathbf{E}}_k|^2, \tag{5}$$

where $|\cdot|$ denotes Euclidean vector norm. Note that the embeddings do not collapse to the trivial solution $\mathbf{E}_0 = \cdots = \mathbf{E}_{p-1} = 0$ unless initialized as such, because two conserved quantities exist, as proven in Appendix F:

$$\mathbf{C} = \sum_k \mathbf{E}_k, \quad Z_0 = \sum_k |\mathbf{E}_k|^2. \tag{6}$$

We shall now use the effective dynamics to explain empirical observations such as the existence of a critical training set size for generalization.

**Degeneracy of ground states (loss optima)** We define ground states as those representations satisfying $\ell_{\mathrm{eff}} = 0$, which requires the following linear equations to hold:

$$A(P) = \{\mathbf{E}_i + \mathbf{E}_j = \mathbf{E}_m + \mathbf{E}_n | (i,j,m,n) \in P\}. \tag{7}$$

Since each embedding dimension obeys the same set of linear equations, we will assume, without loss of generality, that $d_{\mathrm{in}} = 1$. The dimension of the null space of $A(P)$, denoted as $n_0$, is the number of degrees of freedom of the ground states. Given a set of parallelograms implied by a training dataset $D$, the nullity of $A(P(D))$ could be obtained by computing the singular values $0 \le \sigma_1 \le \cdots \le \sigma_p$. We always have $n_0 \ge 2$, i.e., $\sigma_1 = \sigma_2 = 0$ because the nullity of $A(P_0)$, the set of linear equations given by all possible parallelograms, is $\mathrm{Nullity}(A(P_0)) = 2$ which can be attributed to two degrees

of freedom (translation and scaling). If $n_0 = 2$, the representation is unique up to translations and scaling factors, and the embeddings have the form $\mathbf{E}_k = \mathbf{a} + k\mathbf{b}$. Otherwise, when $n_0 > 2$, the representation is not constrained enough such that all the embeddings lie on a line.

We present theoretical predictions alongside empirical results for addition ($p = 10$) in Figure 4. As shown in Figure 4 (a), our effective theory predicts that the probability that the training set implies a unique linear structure (which would result in perfect generalization) depends on the training data fraction and has a phase transition around $r_c = 0.4$. Empirical results from training different models are shown in Figure 4 (b). The number of steps to reach $\mathrm{RQI} > 0.95$ is seen to have a phase transition at $r_c = 0.4$, agreeing with the proposed effective theory and with the empirical findings in [1].

**Time towards the linear structure** We define the Hessian matrix of $\ell_0$ as

$$\mathbf{H}_{ij} = \frac{1}{Z_0} \frac{\partial^2 \ell_0}{\partial \mathbf{E}_i \partial \mathbf{E}_j}, \tag{8}$$

Note that $\ell_{\mathrm{eff}} = \frac{1}{2}\mathbf{R}^T\mathbf{H}\mathbf{R}$, $\mathbf{R} = [\mathbf{E}_0, \mathbf{E}_1, \cdots, \mathbf{E}_{p-1}]$, so the gradient descent is linear, i.e.,

$$\frac{d\mathbf{R}}{dt} = -\mathbf{H}\mathbf{R}. \tag{9}$$

If $\mathbf{H}$ has eigenvalues $\lambda_i = \sigma_i^2$ (sorted in increasing order) and eigenvectors $\bar{\mathbf{v}}_i$, and we have the initial condition $\mathbf{R}(t = 0) = \sum_i a_i\bar{\mathbf{v}}_i$, then we have $\mathbf{R}(t) = \sum_i a_i\bar{\mathbf{v}}_i e^{-\lambda_i t}$. The first two eigenvalues vanish and $t_h = 1/\lambda_3$ determines the timescale for the slowest component to decrease by a factor of $e$. We call $\lambda_3$ the *grokking rate*. When the step size is $\eta$, the corresponding number of steps is $n_h = t_h/\eta = 1/(\lambda_3\eta)$.

We verify the above analysis with empirical results. Figure 4 (c)(d) show the trajectories obtained from the effective theory and from neural network training, respectively. The 1D neural representation in Figure 4 (d) are manually normalized to zero mean and unit variance. The two trajectories agree qualitatively, and it takes about $3n_h$ steps for two trajectories to converge to the linear structure. The quantitative differences might be due to the absence of the decoder in the effective theory, which assumes the decoder to take infinitesimal step sizes.

**Dependence of grokking on data size** Note that $\ell_{\mathrm{eff}}$ involves averaging over parallelograms in the training set, it is dependent on training data size, so is $\lambda_3$. In Figure 5 (a), we plot the dependence of $\lambda_3$ on training data fraction. There are many datasets with the same data size, so $\lambda_3$ is a probabilistic function of data size.

Two insights on grokking can be extracted from this plot: (i) When the data fraction is below some threshold (around 0.4), $\lambda_3$ is zero with high probability, corresponding to no generalization. This again verifies our critical point in Figure 4. (ii) When data size is above the threshold, $\lambda_3$ (on average) is an increasing function of data size. This implies that grokking time $t \sim 1/\lambda_3$ decreases as training data size becomes larger, an important observation from [1].

To verify our effective theory, we compare the grokking steps obtained from real neural network training (defined as steps to $\mathrm{RQI} > 0.95$), and those predicted by our theory $t_{\mathrm{th}} \sim \frac{1}{\lambda_3\eta}$ ($\eta$ is the embedding learning rate), shown in Figure 5 (b). The theory agrees qualitatively with neural networks, showing the trend of decreasing grokking steps as increasing data size. The quantitative differences might be explained as the gap between our effective loss and actual loss.

**Limitations of the effective theory** While our theory defines an effective loss based on the Euclidean distance between embeddings $\mathbf{E}_i + \mathbf{E}_j$ and $\mathbf{E}_n + \mathbf{E}_m$, one could imagine generalizing the theory to define a broader notion of parallogram given by some other metric on the representation space. For instance, if we have a decoder like in Figure 2 (d) then the distance between distinct representations within the same "pizza slice" is low, meaning that representations arranged not in parallelograms w.r.t. the Euclidean metric may be parallelograms with respect to the metric defined by the decoder.

## 4 Delayed Generalization: A Phase Diagram

So far, we have (1) observed empirically that generalization on algorithmic datasets corresponds with the emergence of well-structured representations, (2) defined a notion of representation quality in a toy setting and shown that it predicts generalization, and (3) developed an effective theory to describe

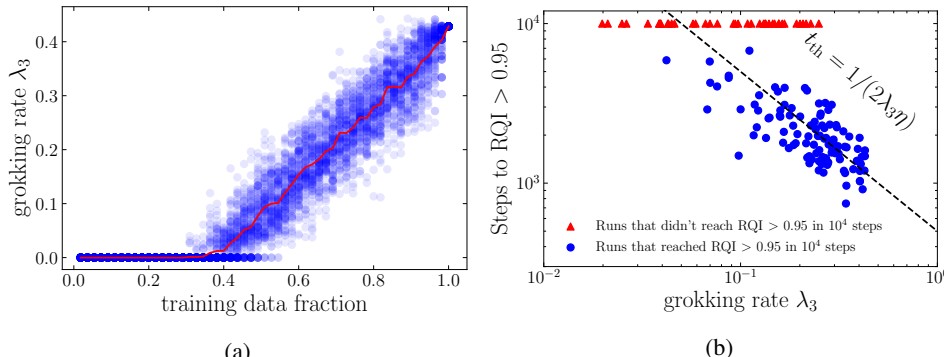

(a)                                                    (b)

Figure 5: Effective theory explains the dependence of grokking time on data size, for the addition task. (a) Dependence of $\lambda_3$ on training data fraction. Above the critical data fraction (around 0.4), as data size becomes larger, $\lambda_3$ increases hence grokking time $t \sim 1/\lambda_3$ (predicted by our effective theory) decreases. (b) Comparing grokking steps (defined as $\mathrm{RQI} > 0.95$) predicted by the effective theory with real neural network results. $\eta = 10^{-3}$ is the learning rate of the embeddings.

the learning dynamics of the representations in the same toy setting. We now study how optimizer hyperparameters affect high-level learning performance. In particular, we develop phase diagrams for how learning performance depends on the representation learning rate, decoder learning rate and the decoder weight decay. These parameters are of interest since they most explicitly regulate a kind of *competition* between the encoder and decoder, as we elaborate below.

## 4.1   Phase diagram of a toy model

**Training details** We update the representation and the decoder with different optimizers. For the 1D embeddings, we use the Adam optimizer with learning rate $[10^{-5}, 10^{-2}]$ and zero weight decay. For the decoder, we use an AdamW optimizer with the learning rate in $[10^{-5}, 10^{-2}]$ and the weight decay in $[0, 10]$ (regression) or $[0, 20]$ (classification). For training/validation spliting, we choose 45/10 for non-modular addition ($p = 10$) and 24/12 for the permutation group $S_3$. We hard-code addition or matrix multiplication (details in Appendix H) in the decoder for the addition group and the permutation group, respectively.

For each choice of learning rate and weight decay, we compute the number of steps to reach high (90%) training/validation accuracy. The 2D plane is split into four phases: *comprehension*, *grokking*, *memorization* and *confusion*, defined in Table 1 in Appendix A. Both comprehension and grokking are able to generalize (in the "Goldilocks zone"), although the grokking phase has delayed generalization. Memorization is also called overfitting, and confusion means failure to even memorize training data. Figure 6 shows the phase diagrams for the addition group and the permutation group. They display quite rich phenomena.

**Competition between representation learning and decoder overfitting** In the regression setup of the addition dataset, we show how the competition between representation learning and decoder learning (which depend on both learning rate and weight decay, among other things) lead to different learning phases in Figure 6 (a). As expected, a fast decoder coupled with slow representation learning (bottom right) lead to memorization. In the opposite extreme, although an extremely slow decoder coupled with fast representation learning (top left) will generalize in the end, the generalization time is long due to the inefficient decoder training. The ideal phase (comprehension) requires representation learning to be faster, but not too much, than the decoder.

Drawing from an analogy to physical systems, one can think of embedding vectors as a group of particles. In our effective theory from Section 3.2, the dynamics of the particles are described *only* by their relative positions, in that sense, structure forms mainly due to inter-particle interactions (in reality, these interactions are mediated by the decoder and the loss). The decoder plays the role of an environment exerting external forces on the embeddings. If the magnitude of the external forces are small/large one can expect better/worse representations.

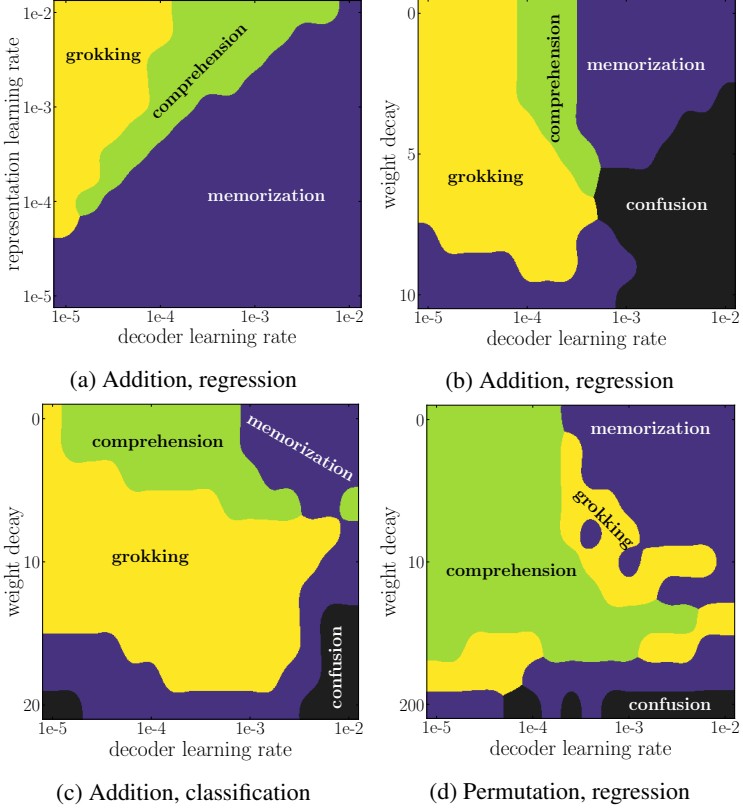

Figure 6: Phase diagrams of learning for the addition group and the permutation group. (a) shows the competition between representation and decoder. (b)(c)(d): each phase diagram contains four phases: comprehension, grokking, memorization and confusion, defined in Table 1. In (b)(c)(d), grokking is sandwiched between comprehension and memorization.

**Universality of phase diagrams** We fix the embedding learning rate to be $10^{-3}$ and sweep instead decoder weight decay in Figure 6 (b)(c)(d). The phase diagrams correspond to addition regression (b), addition classification (c) and permutation regression (d), respectively. Common phenomena emerge from these different tasks: (i) they all include four phases; (ii) The top right corner (a fast and capable decoder) is the memorization phase; (iii) the bottom right corner (a fast and simple decoder) is the confusion phase; (iv) grokking is sandwiched between comprehension and memorization, which seems to imply that it is an undesirable phase that stems from improperly tuned hyperparameters.

## 4.2 Beyond the toy model

We conjecture that many of the principles which we saw dictate the training dynamics in the toy model also apply more generally. Below, we will see how our framework generalizes to transformer architectures for the task of addition modulo $p$, a minimal reproducible example of the original grokking paper [1].

We first encode $p = 53$ integers into 256D learnable embeddings, then pass two integers to a decoder-only transformer architecture. For simplicity, we do not encode the operation symbols here. The outputs from the last layer are concatenated and passed to a linear layer for classification. Training both the encoder and the decoder with the same optimizer (i.e., with the same hyperparameters) leads to the grokking phenomenon. Generalization appears much earlier once we lower the effective decoder capacity with weight decay (full phase diagram in Figure 7).

Early on, the model is able to perfectly fit the training set while having no generalization. We study the embeddings at different training times and find that neither PCA (shown in Figure 1) nor t-SNE (not shown here) reveal any structure. Eventually, validation accuracy starts to increase, and perfect generalization coincides with the PCA projecting the embeddings into a circle in 2D. Of course, no

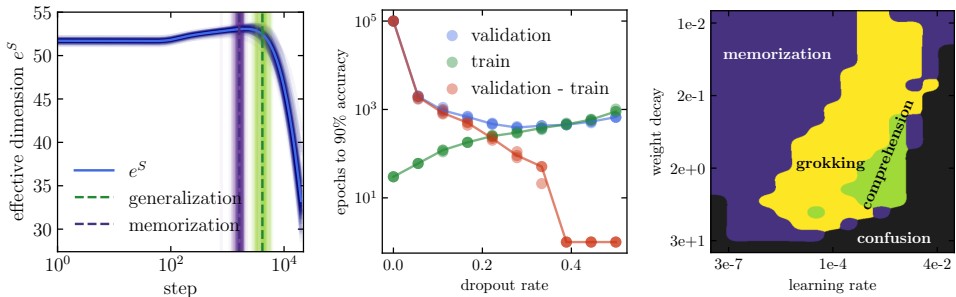

Figure 7: Left: Evolution of the effective dimension of the embeddings (defined as the exponential of the entropy) during training and evaluated over 100 seeds. Center: Effect of dropout on speeding up generalization. Right: Phase diagram of the transformer architecture. A scan is performed over the weight decay and learning rate of the decoder while the learning rate of the embeddings is kept fixed at $10^{-3}$ (with zero weight decay).

choice of dimensionality reduction is guaranteed to find any structure, and thus, it is challenging to show explicitly that generalization only occurs when a structure exists. Nevertheless, the fact that, when coupled with the implicit regularization of the optimizer for sparse solutions, such a clear structure appears in a simple PCA so quickly at generalization time suggests that our analysis in the toy setting is applicable here as well. This is also seen in the evolution of the entropy of the explained variance ratio in the PCA of the embeddings (defined as $S = -\sum_i \sigma_i \log \sigma_i$ where $\sigma_i$ is the fractional variance explained by the $i$th principal component). As seen in Figure 7, the entropy increases up to generalization time then decreases drastically afterwards which would be consistent with the conjecture that generalization occurs when a low-dimensional structure is discovered. The decoder then primarily relies on the information in this low-dimensional manifold and essentially "prunes" the rest of the high-dimensional embedding space. Another interesting insight appears when we project the embeddings at initialization onto the principal axes at the end of training. Some of the structure required for generalization exists before training hinting at a connection with the Lottery Ticket Hypothesis. See Appendix K for more details.

In Figure 7 (right), we show a comparable phase diagram to Figure 6 evaluated now in the transformer setting. Note that, as opposed to the setting in [1], weight decay has only been applied to the decoder and not to the embedding layer. Contrary to the toy model, a certain amount of weight decay proves beneficial to generalization and speeds it up significantly. We conjecture that this difference comes from the different embedding dimensions. With a highly over-parameterized setting, a non-zero weight decay gives a crucial incentive to reduce complexity in the decoder and help generalize in fewer steps. This is subject to further investigation. We also explore the effect of dropout layers in the decoder blocks of the transformer. With a significant dropout rate, the generalization time can be brought down to under $10^3$ steps and the grokking phenomenon vanishes completely. The overall trend suggests that constraining the decoder with the same tools used to avoid overfitting reduces generalization time and can avoid the grokking phenomenon. This is also observed in an image classification task where we were able to induce grokking. See Appendix J for more details.

### 4.3 Grokking Experiment on MNIST

We now demonstrate, for the first time, that grokking (significantly delayed generalization) is a more general phenomenon in machine learning that can occur not only on algorithmic datasets, but also on mainstream benchmark datasets. In particular, we exhibit grokking on MNIST in Figure 8 and demonstrate that we can control grokking by varying optimization hyperparameters. More details on the experimental setup are in Appendix J.

## 5 Related work

Relatively few works have analyzed the phenomenon of grokking. [2] describe the circuit that transformers use to perform modular addition, track its formation over training, and broadly suggest that grokking is related to the phenomenon of "phase changes" in neural network training. [3, 4]

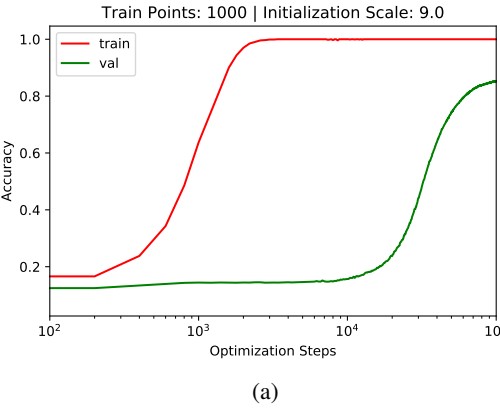 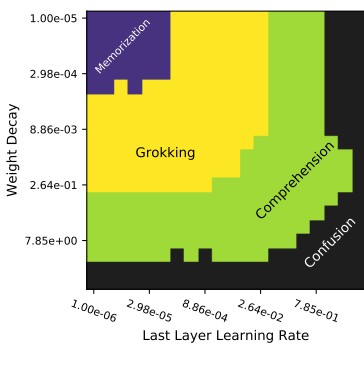

|       |       |
|-------|-------|
| (a)   | (b)   |

Figure 8: Left: Training curves for a run on MNIST, in the setting where we observe grokking. Right: Phase diagram with the four phases of learning dynamics on MNIST.

provided earlier speculative, informal conjectures on grokking [3, 4]. Our work is related to the following broad research directions:

**Learning mathematical structures** [5] trains a neural network to learn arithmetic operation from pictures of digits, but they do not observe grokking due to their abundant training data. Beyond arithmetic relations, machine learning has been applied to learn other mathematical structures, including geometry [6], knot theory [7] and group theory [8].

**Double descent** Grokking is somewhat reminiscent of the phenomena of "epoch-wise" *double descent* [9], where generalization can improve after a period of overfitting. [10] find that regularization can mitigate double descent, similar perhaps to how weight decay influences grokking.

**Representation learning** Representation learning lies at the core of machine learning [11–14]. Representation quality is usually measured by (perhaps vague) semantic meanings or performance on downstream tasks. In our study, the simplicity of arithmetic datasets allows us to define representation quality and study evolution of representations in a quantitative way.

**Physics of learning** Physics-inspired tools have proved to be useful in understanding deep learning from a theoretical perspective. These tools include effective theories [15, 16], conservation laws [17] and free energy principle [18]. In addition, statistical physics has been identified as a powerful tool in studying generalization in neural networks [19–22]. Our work connects a low-level understanding of models with their high-level performance. In a recent work, researchers at Anthropic [23], connect a sudden decrease in loss during training with the emergence of *induction heads* within their models. They analogize their work to *statistical physics*, since it bridges a "microscopic", mechanistic understanding of networks with "macroscopic" facts about overall model performance.

## 6   Conclusion

We have shown how, in both toy models and general settings, that representation enables generalization when it reflects structure in the data. We developed an effective theory of representation learning dynamics (in a toy setting) which predicts the critical dependence of learning on the training data fraction. We then presented four learning phases (comprehension, grokking, memorization and confusion) which depend on the decoder capacity and learning speed (given by, among other things, learning rate and weight decay) in decoder-only architectures. While we have mostly focused on a toy model, we find preliminary evidence that our results generalize to the setting of [1].

Our work can be viewed as a step towards a *statistical physics of deep learning*, connecting the "microphysics" of low-level network dynamics with the "thermodynamics" of high-level model behavior. We view the application of theoretical tools from physics, such as effective theories [24], to be a rich area for further work. The broader impact of such work, if successful, could be to make models more transparent and predictable [23, 25, 26], crucial to the task of ensuring the safety of advanced AI systems.

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
