# Appendix

## A  Definitions of the phases of learning

Table 1: Definitions of the four phases of learning

| Phase | training acc > 90% within $10^5$ steps | validation acc > 90% within $10^5$ steps | step(validation acc>90%) $-$step(training acc>90%)<$10^3$ |
|---|---|---|---|
| | | criteria | |
| **Comprehension** | Yes | Yes | Yes |
| **Grokking** | Yes | Yes | No |
| **Memorization** | Yes | No | Not Applicable |
| **Confusion** | No | No | Not Applicable |

## B  Applicability of our toy setting

In the main paper, we focused on the toy setting with (1) the addition dataset and (2) the addition operation hard coded in the decoder. Although both simplifications appear to have quite limited applicability, we argue below that the analysis of the toy setting can actually apply to all Abelian groups.

**The addition dataset is the building block of all Abelian groups** A cyclic group is a group that is generated by a single element. A finite cyclic group with order $n$ is $C_n = \{e, g, g^2, \cdots, g^{n-1}\}$ where $e$ is the identify element and $g$ is the generator and $g^i = g^j$ whenever $i = j \pmod{n}$. The modulo addition and $\{0, 1, \cdots, n-1\}$ form a cyclic group with $e = 0$ and $g$ can be any number $q$ coprime to $n$ such that $(q, n) = 1$. Since algorithmic datasets contain only symbolic but no arithmetic information, the datasets of modulo addition could apply to all other cyclic groups, e.g., modulo multiplication and discrete rotation groups in 2D.

Although not all Abelian groups are cyclic, a finite Abelian group $G$ can be always decomposed into a direct product of $k$ cyclic groups $G = C_{n_1} \times C_{n_2} \cdots C_{n_k}$. So after training $k$ neural networks with each handling one cyclic group separately, it is easy to construct a larger neural network that handles the whole Abelian group.

**The addition operation is valid for all Abelian groups** It is proved in [27] that for a permutation invariant function $f(x_1, x_2, \cdots, x_n)$, there exists $\rho$ and $\phi$ such that

$$f(x_1, x_2, \cdots, x_n) = \rho[\sum_{i=1}^{n} \phi(x_i)], \tag{10}$$

or $f(x_1, x_2) = \rho(\phi(x_1) + \phi(x_2))$ for $n = 2$. Notice that $\phi(x_i)$ corresponds to the embedding vector $\mathbf{E}_i$, $\rho$ corresponds to the decoder. The addition operator naturally emerges from the commutativity of the operator, not restricting the operator itself to be addition. For example, multiplication of two numbers $x_1$ and $x_2$ can be written as $x_1 x_2 = \exp(\ln(x_1) + \ln(x_2))$ where $\rho(x) = \exp(x)$ and $\phi(x) = \ln(x)$.

## C  An illustrative example

We use a concrete case to illustrate why parallelograms lead to generalization (see Figure 9). For the purpose of illustration, we exploit a curriculum learning setting, where a neural network is fed with a few new samples each time. We will illustrate that, as we have more samples in the training set, the ideal model $\mathcal{M}^*$ (defined in Section 3.2) will arrange the representation $\mathbf{R}^*$ in a more structured way, i.e., more parallelograms are formed, which helps generalization to unseen validation samples. For simplicity we choose $p = 6$.

- $D_1 = (0, 4)$ and $D_2 = (1, 3)$ have the same label, so $(0, 4, 1, 3)$ becomes a parallelogram such that $\mathbf{E}_0 + \mathbf{E}_4 = \mathbf{E}_1 + \mathbf{E}_3 \rightarrow \mathbf{E}_3 - \mathbf{E}_0 = \mathbf{E}_4 - \mathbf{E}_1$. $D_3 = (1, 5)$ and $D_4 = (2, 4)$ have

the same label, so $(1, 5, 2, 4)$ becomes a parallelogram such that $\mathbf{E}_1 + \mathbf{E}_5 = \mathbf{E}_2 + \mathbf{E}_4 \rightarrow \mathbf{E}_4 - \mathbf{E}_1 = \mathbf{E}_5 - \mathbf{E}_2$. We can derive from the first two equations that $\mathbf{E}_5 - \mathbf{E}_2 = \mathbf{E}_3 - \mathbf{E}_0 \rightarrow \mathbf{E}_0 + \mathbf{E}_5 = \mathbf{E}_2 + \mathbf{E}_3$, which implies that $(0, 5, 2, 3)$ is also a parallelogram (see Figure 9(a)). This means if $(0, 5)$ in training set, our model can predict $(2, 3)$ correctly.

- $D_5 = (0, 2)$ and $D_6 = (1, 1)$ have the same label, so $\mathbf{E}_0 + \mathbf{E}_2 = 2\mathbf{E}_1$, i.e., $1$ is the middle point of $0$ and $2$ (see Figure 9(b)). Now we can derive that $2\mathbf{E}_4 = \mathbf{E}_3 + \mathbf{E}_5$, i.e., $4$ is the middle point of $3$ and $5$. If $(4, 4)$ is in the training data, our model can predict $(3, 5)$ correctly.

- Finally, $D_7 = (2, 4)$ and $D_8 = (3, 3)$ have the same label, so $2\mathbf{E}_3 = \mathbf{E}_2 + \mathbf{E}_4$, i.e., $3$ should be placed at the middle point of $2$ and $4$, ending up Figure 9(c). This linear structure agrees with the arithmetic structure of $\mathbb{R}$.

In summary, although we have $p(p + 1)/2 = 21$ different training samples for $p = 6$, we only need $8$ training samples to uniquely determine the perfect linear structure (up to linear transformation). The punchline is: representations lead to generalization.

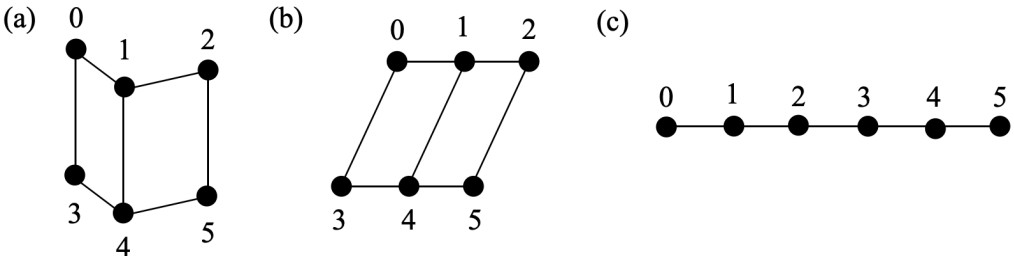

Figure 9: As we include more data in the training set, the (ideal) model is capable of discovering increasingly structured representations (better RQI), from (a) to (b) to (c).

# D   Definition of $\widehat{\text{Acc}}$

Given a training set $D$ and a representation $\mathbf{R}$, if $(i, j)$ is a validation sample, can the neural network correctly predict its output, i.e., $\text{Dec}(\mathbf{E}_i + \mathbf{E}_j) = \mathbf{Y}_{i+j}$? Since neural network has never seen $(i, j)$ in the training set, one possible mechanism of induction is through

$$\text{Dec}(\mathbf{E}_i + \mathbf{E}_j) = \text{Dec}(\mathbf{E}_m + \mathbf{E}_n) = \mathbf{Y}_{m+n}(= \mathbf{Y}_{i+j}). \tag{11}$$

The first equality $\text{Dec}(\mathbf{E}_i + \mathbf{E}_j) = \text{Dec}(\mathbf{E}_m + \mathbf{E}_n)$ holds only when $\mathbf{E}_i + \mathbf{E}_j = \mathbf{E}_m + \mathbf{E}_n$ (i.e., $(i, j, m, n)$ is a parallelogram). The second equality $\text{Dec}(\mathbf{E}_m + \mathbf{E}_n) = \mathbf{Y}_{m+n}$, holds when $(m, n)$ is in the training set, i.e., $(m, n) \in D$, under the zero training loss assumption. Rigorously, given a training set $D$ and a parallelogram set $P$ (which can be calculated from $\mathbf{R}$), we collect all zero loss samples in an *augmented* training set $\overline{D}$

$$\overline{D}(D, P) = D \bigcup \{(i, j) | \exists (m, n) \in D, (i, j, m, n) \in P\}. \tag{12}$$

Keeping $D$ fixed, a larger $P$ would probably produce a larger $\overline{D}$, i.e., if $P_1 \subseteq P_2$, then $\overline{D}(D, P_1) \subseteq \overline{D}(P, P_2)$, which is why in Eq. (3) our defined RQI $\propto |P|$ gets its name "representation quality index", because higher RQI normally means better generalization. Finally, the expected accuracy from a dataset $D$ and a parallelogram set $P$ is:

$$\widehat{\text{Acc}} = \frac{|\overline{D}(D, P)|}{|D_0|}, \tag{13}$$

which is the estimated accuracy (of the full dataset), and $P = P(\mathbf{R})$ is defined on the representation after training. On the other hand, accuracy Acc can be accessed empirically from trained neural network. We verified Acc $\approx \widehat{\text{Acc}}$ in a toy setup (addition dataset $p = 10$, 1D embedding space, hard code addition), as shown in Figure 3 (c). Figure 3 (a)(b) show Acc and $\widehat{\text{Acc}}$ as a function of training set ratio, with each dot corresponding to a different random seed. The dashed red diagonal corresponds to memorization of the training set, and the vertical gap refers to generalization.

Although the agreement is good for 1D embedding vectors, we do not expect such agreement can trivially extend to high dimensional embedding vectors. In high dimensions, our definition of RQI is too restrictive. For example, suppose we have an embedding space with $N$ dimensions. Although the representation may form a linear structure in the first dimension, the representation can be arbitrary in other $N - 1$ dimensions, leading to RQI $\approx 0$. However, the model may still generalize well if the decoder learns to keep only the useful dimension and drop all other $N - 1$ useless dimensions. It would be interesting to investigate how to define an RQI that takes into account the role of decoder in future works.

## E    The gap of a realistic model $\mathcal{M}$ and the ideal model $\mathcal{M}^*$

Realistic models $\mathcal{M}$ usually form fewer number of parallelograms than ideal models $\mathcal{M}^*$. In this section, we analyze the properties of ideal models and calculated ideal RQI and ideal accuracy, which set upper bounds for empirical RQI and accuracy. The upper bound relations are verified via numerical experiments in Figure 10.

Similar to Eq. (12) where some validation samples can be derived from training samples, we demonstrate how *implicit parallelograms* can be 'derived' from explicit ones in $P_0(D)$. The so-called derivation follows a simple geometric argument that: if $A_1B_1$ is equal and parallel to $A_2B_2$, and $A_2B_2$ is equal and parallel to $A_3B_3$, then we can deduce that $A_1B_1$ is equal and parallel to $A_3B_3$ (hence $(A_1, B_2, A_2, B_1)$ is a parallelogram).

Recall that a parallelogram $(i, j, m, n)$ is equivalent to $\mathbf{E}_i + \mathbf{E}_j = \mathbf{E}_m + \mathbf{E}_n$ ($*$). So we are equivalently asking if equation ($*$) can be expressed as a linear combination of equations in $A(P_0(D))$. If yes, then ($*$) is dependent on $A(P_0(D))$ (defined in Eq. (7)), i.e., $A(P_0(D))$ and $A(P_0(D) \bigcup (i, j, m, n))$ should have the same rank. We augment $P_0(D)$ by adding implicit parallelograms, and denote the augmented parallelogram set as

$$P(D) = P_0(D) \bigcup \{q \equiv (i, j, m, n) | q \in P_0, \mathrm{rank}(A(P_0(D))) = \mathrm{rank}(A(P_0(D) \bigcup q))\}. \quad (14)$$

We need to emphasize that an assumption behind Eq. (14) is that we have an ideal model $\mathcal{M}^*$. When the model is not ideal, e.g., when the injectivity of the encoder breaks down, fewer parallelograms are expected to form, i.e.,

$$P(R) \subseteq P(D). \quad (15)$$

The inequality is saying, whenever a parallelogram is formed in the representation after training, the reason is hidden in the training set. This is not a strict argument, but rather a belief that today's neural networks can only copy what datasets (explicitly or implicitly) tell it to do, without any autonomous creativity or intelligence. For simplicity we call this belief *Alexander Principle*. In very rare cases when something lucky happens (e.g., neural networks are initialized at approximate correct weights), Alexander principle may be violated. Alexander principle sets an upper bound for RQI:

$$\mathrm{RQI}(R) \leq \frac{|P(D)|}{|P_0|} \equiv \overline{\mathrm{RQI}}, \quad (16)$$

and sets an upper bound for $\widehat{\mathrm{Acc}}$:

$$\widehat{\mathrm{Acc}} \equiv \widehat{\mathrm{Acc}}(D, P(R)) \leq \widehat{\mathrm{Acc}}(D, P(D)) \equiv \overline{\mathrm{Acc}}. \quad (17)$$

In Figure 10 (c)(f), we verify Eq. (16) and Eq. (17). We choose $\delta = 0.01$ to compute RQI($R,\delta$). We find the trained models are usually far from being ideal, although we already include a few useful tricks proposed in Section 4 to enhance representation learning. It would be an interesting future direction to develop better algorithms so that the gap due to Alexander principle can be reduced or even closed. In Figure 10 (a)(b)(d)(e), four quantities (RQI, $\overline{\mathrm{RQI}}$, Acc, $\overline{\mathrm{Acc}}$) as functions of the training data fraction are shown, each dot corresponding to one random seed. It is interesting to note that it is possible to have $\overline{\mathrm{RQI}} = 1$ only with $< 40\%$ training data, i.e., $55 \times 0.4 = 22$ samples, agreeing with our observation in Section 3.

**Realistic representations** Suppose an ideal model $\mathcal{M}^*$ and a realistic model $\mathcal{M}$ which train on the training set $D$ give the representation $R^*$ and $R$, respectively. What is the relationship between $R$ and $R_*$? Due to the Alexander principle we know $P(R) \subseteq P(D) = P(R^*)$. This means $R^*$ has more parallelograms than $R$, hence $R^*$ has fewer degrees of freedom than $R$.

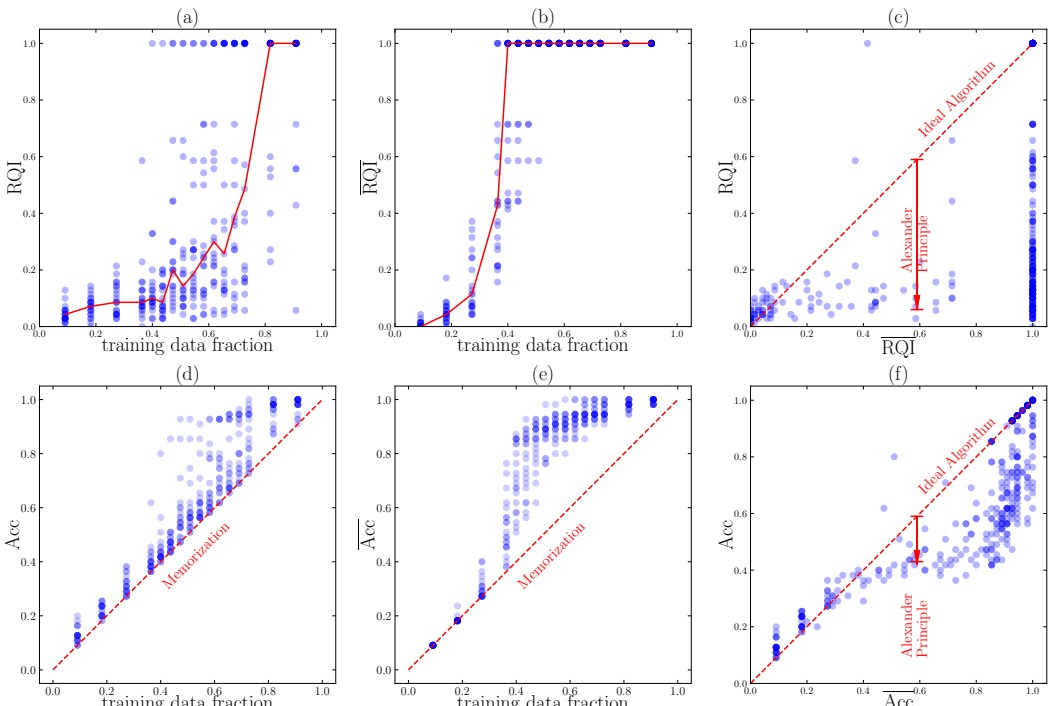

Figure 10: We compare RQI and Acc for an ideal algorithm (with bar) and a realistic algorithm (without bar). In (a)(b)(d)(e), four quantities (RQI, $\overline{\text{RQI}}$, Acc, $\overline{\text{Acc}}$) as functions of training data fraction are shown. In (c)(f), RQI and Acc of the ideal algorithm sets upper bounds for those of the realistic algorithm.

We illustrate with the toy case $p = 4$. The whole dataset contains $p(p + 1)/2 = 10$ samples, i.e.,

$$D_0 = \{(0,0), (0,1), (0,2), (0,3), (1,1), (1,2), (1,3), (2,2), (2,3), (3,3)\}. \tag{18}$$

The parallelogram set contains only three elements, i.e.,

$$P_0 = \{(0,1,1,2), (0,1,2,3), (1,2,2,3)\}, \tag{19}$$

Or equivalently the equation set

$$A_0 = \{\text{A1} : \mathbf{E}_0 + \mathbf{E}_2 = 2\mathbf{E}_1, \text{A2} : \mathbf{E}_0 + \mathbf{E}_3 = \mathbf{E}_1 + \mathbf{E}_2, \text{A3} : \mathbf{E}_1 + \mathbf{E}_3 = 2\mathbf{E}_2\}. \tag{20}$$

Pictorially, we can split all possible subsets $\{A|A \subseteq A_0\}$ into different levels, each level defined by $|A|$ (the number of elements). A subset $A_1$ in the $i^{\text{th}}$ level points an direct arrow to another subset $A_2$ in the $(i + 1)^{\text{th}}$ level if $A_2 \subset A_1$, and we say $A_2$ is a child of $A_1$, and $A_1$ is a parent of $A_2$. Each subset $A$ can determine a representation $R$ with $n(A)$ degrees of freedom. So $R$ should be a descendant of $R_*$, and $n(R_*) \leq n(R)$. Numerically, $n(A)$ is equal to the dimension of the null space of $A$.

Suppose we have a training set

$$D = \{(0,2), (1,1), (0,3), (1,2), (1,3), (2,2)\}, \tag{21}$$

and correspondingly $P(D) = P_0, A(P) = A_0$. So an ideal model $\mathcal{M}_*$ will have the linear structure $\mathbf{E}_k = \mathbf{a} + k\mathbf{b}$ (see Figure 11 leftmost). However, a realistic model $\mathcal{M}$ may produce any descendants of the linear structure, depending on various hyperparameters and even random seeds.

In Figure 12, we show our algorithms actually generates all possible representations. We have two settings: (1) fast decoder $(\eta_1, \eta_2) = (10^{-3}, 10^{-2})$ (Figure 12 left), and (2) relatively slow decoder $(\eta_1, \eta_2) = (10^{-2}, 10^{-3})$ (Figure 12 right). The relatively slow decoder produces better representations (in the sense of higher RQI) than a fast decoder, agreeing with our observation in Section 4.

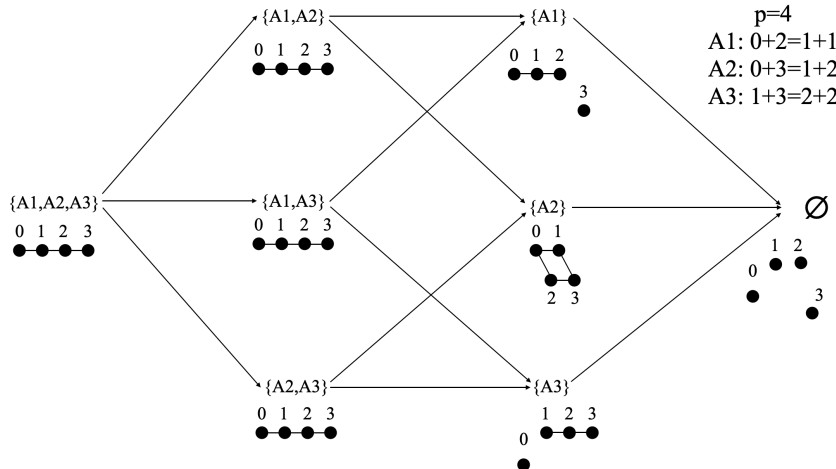

Figure 11: $p = 4$ case. Equation set $A$ (or geometrically, representation) has a hierarchy: $a \to b$ means $a$ is a parent of $b$, and $b$ is a child of $a$. A realistic model can only generate representations that are descendants of the representation generated by an ideal model.

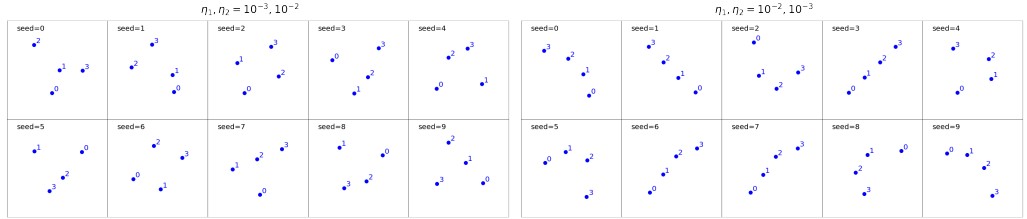

Figure 12: $p = 4$ case. Representations obtained from training neural networks are displayed. $\eta_1$ and $\eta_2$ are learning rates of the representation and the decoder, respectively. As described in the main text, $(\eta_1, \eta_2) = (10^{-2}, 10^{-3})$ (right) is more ideal than $(\eta_1, \eta_2) = (10^{-3}, 10^{-2})$ (left), thus producing representations containing more parallelograms.

## F    Conservation laws of the effective theory

Recall that the effective loss function

$$\ell_{\text{eff}} = \frac{\ell_0}{Z_0}, \quad \ell_0 \equiv \sum_{(i,j,m,n) \in P_0(D)} |\mathbf{E}_i + \mathbf{E}_j - \mathbf{E}_m - \mathbf{E}_n|^2 / |P_0(D)|, \quad Z_0 \equiv \sum_k |\mathbf{E}_k|^2 \quad (22)$$

where $\ell_0$ and $Z_0$ are both quadratic functions of $R = \{\mathbf{E}_0, \cdots, \mathbf{E}_{p-1}\}$, and $\ell_{\text{eff}} = 0$ remains zero under rescaling and translation $\mathbf{E}'_i = a\mathbf{E}_i + \mathbf{b}$. We will ignore the $1/|P_0(D)|$ factor in $\ell_0$ since having it is equivalent to rescaling time, which does not affect conservation laws. The representation vector $\mathbf{E}_i$ evolves according to the gradient descent

$$\frac{d\mathbf{E}_i}{dt} = -\frac{\partial \ell_{\text{eff}}}{\partial \mathbf{E}_i}. \quad (23)$$

We will prove the following two quantities are conserved:

$$\mathbf{C} = \sum_k \mathbf{E}_k, \quad Z_0 = \sum_k |\mathbf{E}_k|^2. \quad (24)$$

Eq. (22) and Eq. (23) give

$$\frac{d\mathbf{E}_i}{dt} = -\frac{\ell_{\text{eff}}}{\partial \mathbf{E}_i} = -\frac{\partial(\frac{\ell_0}{Z_0})}{\partial \mathbf{E}_i} = -\frac{1}{Z_0}\frac{\partial \ell_0}{\partial \mathbf{E}_i} + \frac{\ell_0}{Z_0^2}\frac{\partial Z_0}{\partial \mathbf{E}_i}. \quad (25)$$

Then

$$\frac{dZ_0}{dt} = 2 \sum_i \mathbf{E}_k \cdot \frac{d\mathbf{E}_k}{dt} \tag{26}$$

$$= \frac{2}{Z_0^2} \sum_i \mathbf{E}_i \cdot (-Z_0 \frac{\partial \ell_0}{\partial \mathbf{E}_k} + 2\ell_0 \mathbf{E}_k)$$

$$= \frac{2}{Z_0} (- \sum_k \frac{\partial \ell_0}{\partial \mathbf{E}_k} \cdot \mathbf{E}_k + 2\ell_0)$$

$$= 0.$$

where the last equation uses the fact that

$$\sum_k \frac{\partial \ell_0}{\partial \mathbf{E}_k} \cdot \mathbf{E}_k = 2 \sum_k \sum_{(i,j,m,n) \in P_0(D)} (\mathbf{E}_i + \mathbf{E}_j - \mathbf{E}_m - \mathbf{E}_n)(\delta_{ik} + \delta_{jk} - \delta_{mk} - \delta_{nk}) \cdot \mathbf{E}_k$$

$$= 2 \sum_{(i,j,m,n) \in P_0(D)} (\mathbf{E}_i + \mathbf{E}_j - \mathbf{E}_m - \mathbf{E}_n) \sum_k (\delta_{ik} + \delta_{jk} - \delta_{mk} - \delta_{nk}) \cdot \mathbf{E}_k$$

$$= \sum_{(i,j,m,n) \in P_0(D)} (\mathbf{E}_i + \mathbf{E}_j - \mathbf{E}_m - \mathbf{E}_n) \cdot (\mathbf{E}_i + \mathbf{E}_j - \mathbf{E}_m - \mathbf{E}_n)$$

$$= 2\ell_0$$

The conservation of $Z_0$ prohibits the representation from collapsing to zero. Now that we have demonstrated that $Z_0$ is a conserved quantity, we can also show

$$\frac{d\mathbf{C}}{dt} = \sum_k \frac{d\mathbf{E}_k}{dt} \tag{27}$$

$$= -\frac{1}{Z_0} \sum_k \frac{\partial \ell_0}{\partial \mathbf{E}_k}$$

$$= -\frac{2}{Z_0} \sum_k \sum_{(i,j,m,n) \in P_0(D)} (\mathbf{E}_i + \mathbf{E}_j - \mathbf{E}_m - \mathbf{E}_n)(\delta_{ik} + \delta_{jk} - \delta_{mk} - \delta_{nk})$$

$$= \mathbf{0}.$$

The last equality holds because the two summations can be swapped and $\sum_k (\delta_{ik} + \delta_{jk} - \delta_{mk} - \delta_{nk}) = 0$.

## G    More phase diagrams of the toy setup

We study another three hyperparameters in the toy setup by showing phase diagrams similar to Figure 6. The toy setup is: (1) addition without modulo ($p = 10$); (2) training/validation is split into 45/10; (3) hard code addition; (4) 1D embedding. In the following experiments, the decoder is an MLP with size 1-200-200-30. The representation and the encoder are optimized with AdamW with different hyperparameters. The learning rate of the representation is $10^{-3}$. We sweep the learning rate of the decoder in range $[10^{-4}, 10^{-2}]$ as the x axis, and sweep another hyperparameter as the y axis. By default, we use full batch size 45, initialization scale $s = 1$ and zero weight decay of representation.

**Batch size** controls the amount of noise in the training dynamics. In Figure 13, the grokking region appears at the top left of the phase diagram (small decoder learning rate and small batch size). However, large batch size (with small learning rate) leads to comprehension, implying that smaller batch size seems harmful. This makes sense since to get crystals (good structures) in experiments, one needs a freezer which gradually decreases temperature, rather than something perturbing the system with noise.

**Initialization scale** controls distances among embedding vectors at initialization. We initialize components of embedding vectors from independent uniform distribution $U[-s/2, s/2]$ where $s$

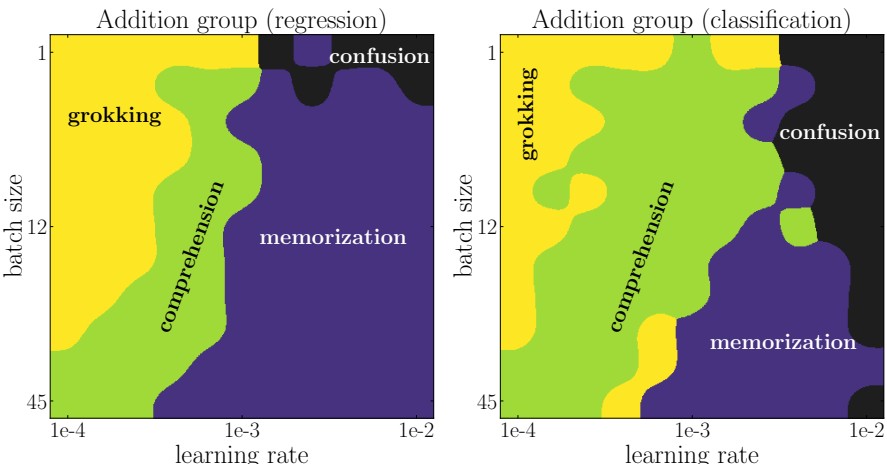

Figure 13: Phase diagrams of decoder learning rate (x axis) and batch size (y axis) for the addition group (left: regression; right: classification). Small decoder leanrning rate and large batch size (bottom left) lead to comprehension.

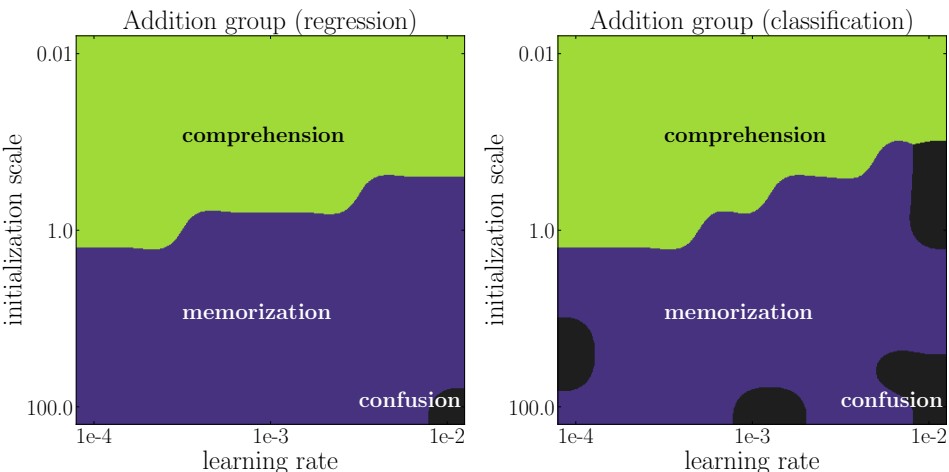

Figure 14: Phase diagrams of decoder learning rate (x axis) and initialization (y axis) for the addition group (left: regression; right: classification). Small intialization scale (top) leads to comprehension.

is called the initialization scale. Shown in Figure 14, it is beneficial to use a smaller initialization scale. This agrees with the physical intuition that closer particles are more likely to interact and form structures. For example, the distances among molecules in ice are much smaller than distances in gas.

**Representation weight decay** controls the magnitude of embedding vectors. Shown in Figure 15, we see the representation weight decay in general does not affect model performance much.

# H  General groups

## H.1  Theory

We focused on Abelian groups for the most part of the paper. This is, however, simply due to pedagogical reasons. In this section, we show that it is straight-forward to extend definitions of parallelograms and representation quality index (RQI) to general non-Abelian groups. We will also show that most (if not all) qualitative results for the addition group also apply to the permutation group.

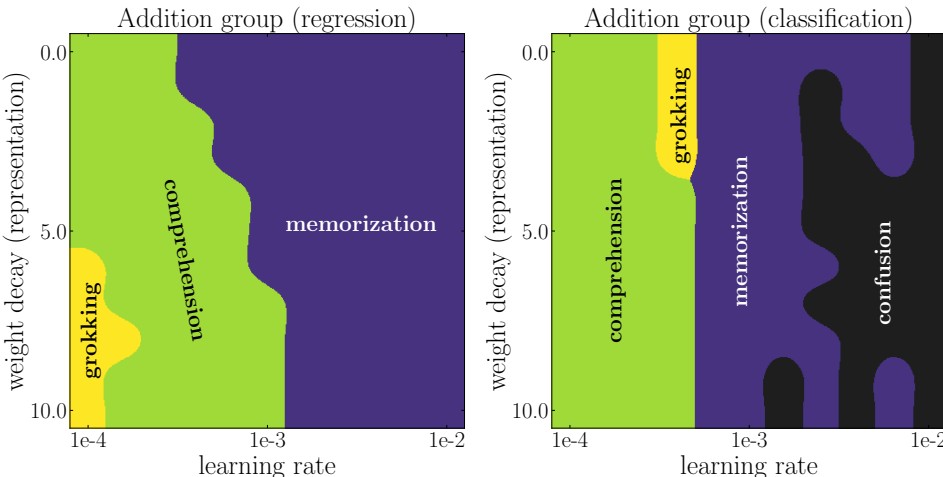

Figure 15: Phase diagrams of decoder learning rate (x axis) and representation weight decay (y axis) for the addition group (left: regression; right: classification). Representation weight decay does not affect model performance much.

**Matrix representation for general groups** Let us first review the definition of group representation. A representation of a group $G$ on a vector space $V$ is a group homomorphism from $G$ to $\mathrm{GL}(V)$, the general linear group on $V$. That is, a representation is a map $\rho : G \to \mathrm{GL}(V)$ such that

$$\rho(g_1 g_2) = \rho(g_1)\rho(g_2), \quad \forall g_1, g_2 \in G. \tag{28}$$

In the case $V$ is of finite dimension $n$, it is common to identify $\mathrm{GL}(V)$ with $n$ by $n$ invertible matrices. The punchline is that: each group element can be represented as a matrix, and the binary operation is represented as matrix multiplication.

**A new architecture for general groups** Inspired by the matrix representation, we embed each group element $a$ as a learnable matrix $\mathbf{E}_a \in \mathbb{R}^{d \times d}$ (as opposed to a vector), and manually do matrix multiplication before sending the product to the deocder for regression or classification. More concretely, for $a \circ b = c$, our architecture takes as input two embedding matrices $\mathbf{E}_a$ and $\mathbf{E}_b$ and aims to predict $\mathbf{Y}_c$ such that $\mathbf{Y}_c = \mathrm{Dec}(\mathbf{E}_a \mathbf{E}_b)$, where $\mathbf{E}_a \mathbf{E}_b$ means the matrix multiplication of $\mathbf{E}_a$ and $\mathbf{E}_b$. The goal of this simplication is to disentangle learning the representation and learning the arithmetic operation (i.e, the matrix multiplication). We will show that, even with this simplification, we are still able to reproduce the characteristic grokking behavior and other rich phenomenon.

**Generalized parallelograms** we define generalized parallelograms: $(a, b, c, d)$ is a generalized parallelogram in the representation if $||\mathbf{E}_a \mathbf{E}_b - \mathbf{E}_c \mathbf{E}_d||_F^2 \leq \delta$, where $\delta > 0$ is a threshold to tolerate numerical errors. Before presenting the numerical results for the permutation group, we show an intuitive picture about how new parallelograms can be deduced from old ones for general groups, which is the key to generalization.

**Deduction of parallelograms** We first recall the case of the Abelian group (e.g., addition group). As shown in Figure 16, when $(a, d, b, c)$ and $(c, f, d, e)$ are two parallelograms, we have

$$\begin{aligned} \mathbf{E}_a + \mathbf{E}_d &= \mathbf{E}_b + \mathbf{E}_c, \\ \mathbf{E}_c + \mathbf{E}_f &= \mathbf{E}_d + \mathbf{E}_d. \end{aligned} \tag{29}$$

We can derive that $\mathbf{E}_a + \mathbf{E}_f = \mathbf{E}_b + \mathbf{E}_e$ implying that $(a, f, b, e)$ is also a parallelogram. That is, for Abelian groups, two parallelograms are needed to deduce a new parallelogram.

For the non-Abelian group, if we have only two parallelograms such that

$$\begin{aligned} \mathbf{E}_a \mathbf{E}_d &= \mathbf{E}_b \mathbf{E}_c, \\ \mathbf{E}_f \mathbf{E}_c &= \mathbf{E}_e \mathbf{E}_d, \end{aligned} \tag{30}$$

we have $\mathbf{E}_b^{-1} \mathbf{E}_a = \mathbf{E}_c \mathbf{E}_d^{-1} = \mathbf{E}_f^{-1} \mathbf{E}_e$, but this does not lead to something like $\mathbf{E}_f \mathbf{E}_a = \mathbf{E}_e \mathbf{E}_b$, hence useless for generalization. However, if we have a third parallelogram such that

$$\mathbf{E}_e \mathbf{E}_h = \mathbf{E}_f \mathbf{E}_g \tag{31}$$

(a) Abelian Group: $2P \rightarrow P$

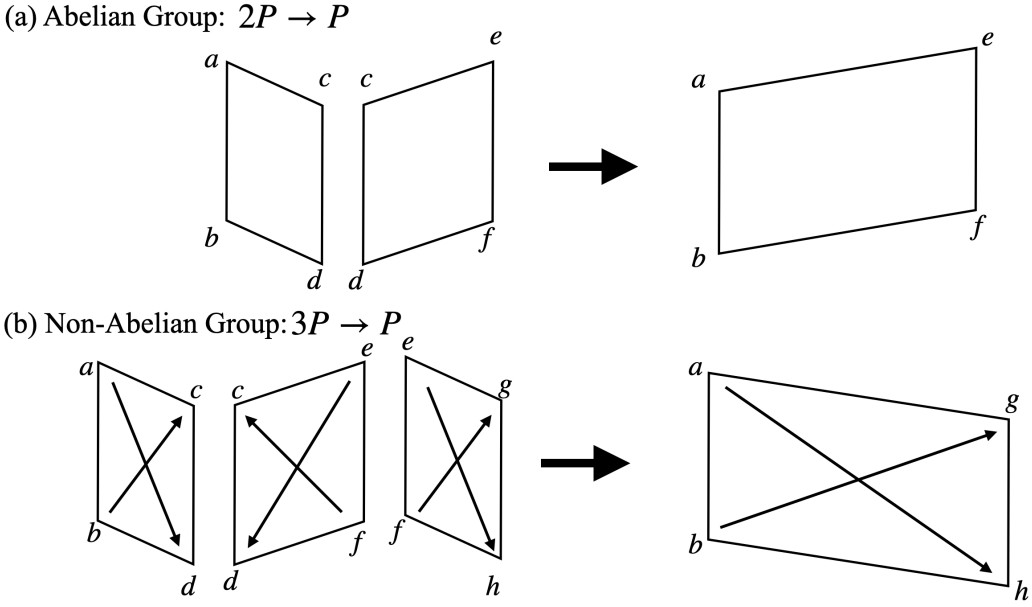

(b) Non-Abelian Group: $3P \rightarrow P$

Figure 16: Deduction of parallelograms

we have $\mathbf{E}_b^{-1}\mathbf{E}_a = \mathbf{E}_c\mathbf{E}_d^{-1} = \mathbf{E}_f^{-1}\mathbf{E}_e = \mathbf{E}_g\mathbf{E}_h^{-1}$, equivalent to $\mathbf{E}_a\mathbf{E}_h = \mathbf{E}_b\mathbf{E}_g$, thus establishing a new parallelogram $(a, h, b, g)$. That is, for non-Abelian groups, three parallelograms are needed to deduce a new parallelogram.

## H.2 Numerical Results

In this section, we conduct numerical experiments on a simple non-abelian group: the permutation group $S_3$. The group has 6 group elements, hence the full dataset contains 36 samples. We embed each group element $a$ into a learnable $3 \times 3$ embedding matrix $\mathbf{E}_a$. We adopt the new architecture described in the above subsection: we hard code matrix multiplication of two input embedding matrices before feeding to the decoder. After defining the generalized parallelogram in the last subsection, we can continue to define RQI (as in Section 3) and predict accuracy $\widehat{\mathrm{Acc}}$ from representation (as in appendix D). We also compute the number of steps needed to reach RQI = 0.95.

**Representation** We flatten each embedding matrix into a vector, and apply principal component analysis (PCA) to the vectors. We show the first three principal components of these group elements in Figure 17. On the plane of PC1 and PC3, the six points are organized as a hexagon.

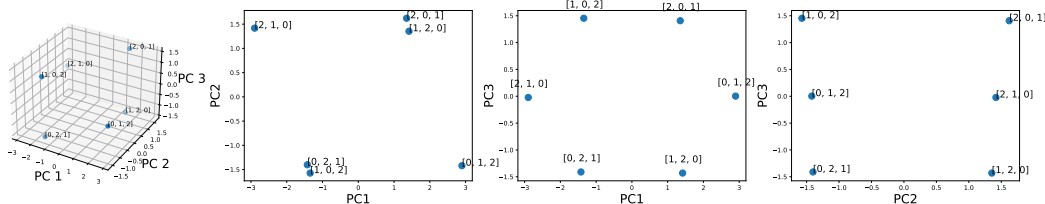

Figure 17: Permuation group $S_3$. First three principal components of six embedding matrices $\mathbb{R}^{3 \times 3}$.

**RQI** In Figure 18 (a), we show RQI as a function of training data fraction. For each training data fraction, we run 11 random seeds (shown as scatter points), and the blue line corresponds to the highest RQI.

**Steps to reach RQI**= 0.95 In Figure 18 (b), we whow the steps to reach RQI > 0.95 as a function of training data fraction, and find a phase transition at $r = r_c = 0.5$. The blue line corresponds to the best model (smallest number of steps).

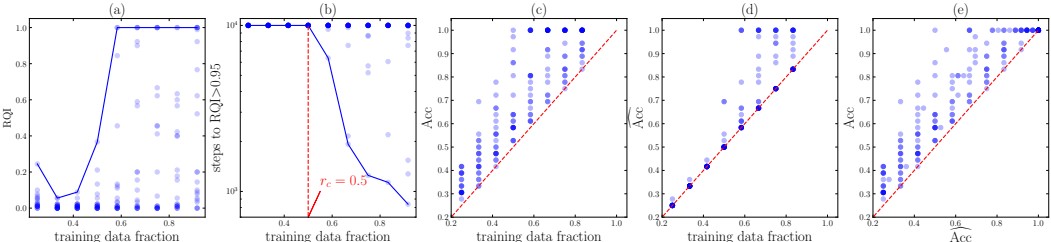

Figure 18: Permutation group $S_3$. (a) RQI increases as training set becomes larger. Each scatter point is a random seed, and the blue line is the highest RQI obtained with a fixed training set ratio; (b) steps to reach $\mathrm{RQI} > 0.95$. The blue line is the smallest number of steps required. There is a phase transition around $r_c = 0.5$. (c) real accuracy Acc; (d) predicted accuracy $\widehat{\mathrm{Acc}}$; (e) comparison of Acc and $\widehat{\mathrm{Acc}}$: $\widehat{\mathrm{Acc}}$ serves as a lower bound of Acc.

**Accuracy** The real accuracy Acc is shown in Figure 18 (c), while the predicted accuracy $\widehat{\mathrm{Acc}}$ (calculated from RQI) is shown in Figure 18 (d). Their comparison is shown in (e): $\widehat{\mathrm{Acc}}$ is a lower bound of Acc, implying that there must be some generalization mechanism beyond RQI.

**Phase diagram** We investigate how the model performance varies under the change of two knobs: decoder learning rate and decoder weight decay. We calculate the number of steps to training accuracy $\geq 0.9$ and validation accuracy $\geq 0.9$, respectively, shown in Figure 6 (d).

## I   Effective theory for image classification

In this section, we show our effective theory proposed in Section 3.2 can generalize beyond algorithmic datasets. In particular, we will apply the effective theory to image classifications. We find that: (i) The effective theory naturally gives rise to a novel self-supervised learning method, which can provably avoid mode collapse without contrastive pairs. (ii) The effective theory can shed light on the neural collapse phenomenon [28], in which same-class representations collapse to their class-means.

We first describe how the effective theory applies to image classification. The basic idea is again that, similar to algorithmic datasets, neural networks try to develop a structured representation of the inputs based on the relational information between samples (class labels in the case of image classification, sum parallelograms in the case of addition, etc.). The effective theory has two ingredients: (i) samples with the same label are encouraged to have similar representations; (ii) the effective loss function is scale-invariant to avoid all representations collapsing to zero (global collapse). As a result, an effective loss for image classification has the form

$$\ell_{\mathrm{eff}} = \frac{\ell}{Z}, \quad \ell = \sum_{(\mathbf{x}, \mathbf{y}) \in P} |\mathbf{f}(\mathbf{x}) - \mathbf{f}(\mathbf{y})|^2, \quad Z = \sum_{\mathbf{x}} |\mathbf{f}(\mathbf{x})|^2 \tag{32}$$

where $\mathbf{x}$ is an image, $\mathbf{f}(\mathbf{x})$ is its representation, $(\mathbf{x}, \mathbf{y}) \in P$ refers to unique pairs $\mathbf{x}$ and $\mathbf{y}$ that have the same label. Scale invariance means the loss function $\ell_{\mathrm{eff}}$ does not change under the linear scaling $\mathbf{f}(\mathbf{x}) \to a\mathbf{f}(\mathbf{x})$.

**Relation to neural collapse** It was observed in [28] that image representations in the penultimate layer of the model have some interesting features: (i) representations of same-class images collapse to their class-means; (ii) class-means of different classes develop into an equiangular tight frame. Our effective theory is able to predict the same-class collapse, but does not necessarily put class-means into equiangular tight frames. We conjecture that little explicit repulsion among different classes can help class-means develop into an equiangular tight frame, similar to electrons developing into lattice structures on a sphere under repulsive Coulomb forces (the Thomson problem [29]). We would like to investigate this modification of the effective theory in the future.

**Experiment on MNIST** We directly apply the effective loss Eq. (32) to the MNIST dataset. Firstly, each image $\mathbf{x}$ is randomly encoded to a 2D embedding $\mathbf{f}(\mathbf{x})$ via the same encoder MLP whose weights are randomly initialized. We then train these embeddings by minimizing the effective loss $\ell_{\mathrm{eff}}$ with

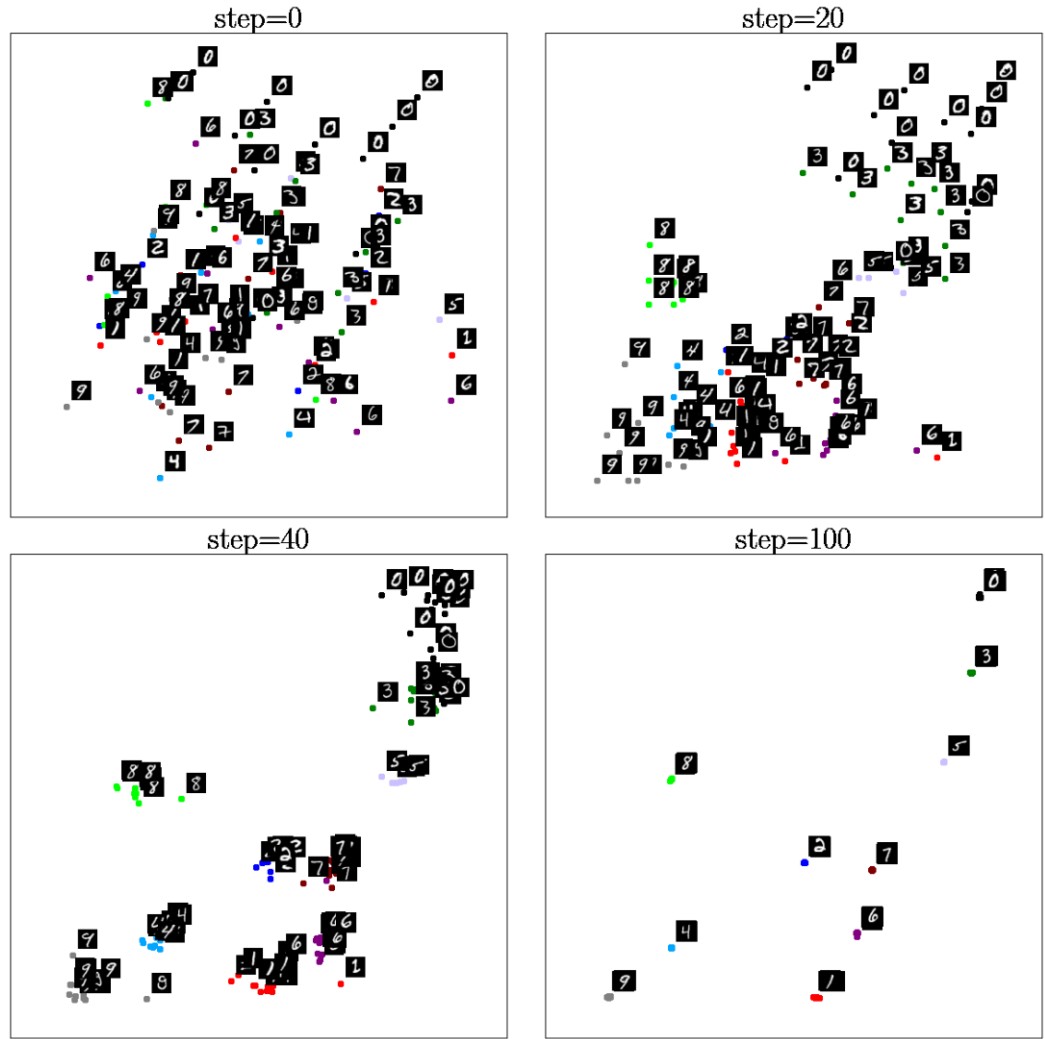

Figure 19: Our effective theory applies to MNIST image classifications. Same-class images collapse to their class-means, while class-means of different classes stay separable. As such, the effective theory serves as a novel self-supervised learning method, as well as shed some light on neural collapse. Please see texts in Appendix I.

an Adam optimizer ($10^{-3}$ learning rate) for 100 steps. We show the evolution of these embeddings in Figure 19. Images of the same class collapse to their class-means, and different class-means do not collapse. This means that our effective theory can give rise to a good representation learning method which only exploits non-contrastive relational information in datasets.

**Link to self-supervised learning** Note that $\ell$ itself is vulnerable to global collapse, in the context of Siamese learning without contrastive pairs. Various tricks (e.g., decoder with momentum, stop gradient) [13, 30] have been proposed to avoid global collapse. However, the reasons why these tricks can avoid global collapse are unclear. We argue $\ell$ fails simply because $\ell \to a^2\ell$ under scaling $\mathbf{f}(\mathbf{x}) \to a\mathbf{f}(\mathbf{x})$ so gradient descent on $\ell$ encourage $a \to 0$. Based on this picture, our effective theory provides another possible fix: make the loss function $\ell$ scale-invariant (by the normalized loss $\ell_{\text{eff}}$), so the gradient flow has no incentive to change representation scales. In fact, we can prove that the gradient flow on $\ell_{\text{eff}}$ preserve $Z$ (variance of representations) so that global collapse is avoided provably:

$$\frac{\partial \ell_{\text{eff}}}{\partial \mathbf{f}(\mathbf{x})} = \frac{1}{Z} \frac{\partial \ell}{\partial \mathbf{f}(\mathbf{x})} - \frac{l}{Z^2} \frac{\partial Z}{\partial \mathbf{f}(\mathbf{x})} = \frac{2}{Z} \sum_{\mathbf{y} \sim \mathbf{x}} (\mathbf{f}(\mathbf{x}) - \mathbf{f}(\mathbf{y})) - \frac{2\ell}{Z^2} \mathbf{f}(\mathbf{x}),$$

$$\frac{dZ}{dt} = 2 \sum_{\mathbf{x}} \mathbf{f}(\mathbf{x}) \cdot \frac{d\mathbf{f}(\mathbf{x})}{dt} = 2 \sum_{\mathbf{x}} \mathbf{f}(\mathbf{x}) \cdot \frac{\partial \ell_{\text{eff}}}{\partial \mathbf{f}(\mathbf{x})}$$

$$= \frac{4}{Z} \sum_{\mathbf{x}} \mathbf{f}(\mathbf{x}) \cdot \left( \sum_{\mathbf{y} \sim \mathbf{x}} (\mathbf{f}(\mathbf{x}) - \mathbf{f}(\mathbf{y})) - \frac{\ell}{Z} \mathbf{f}(\mathbf{x}) \right) \qquad (33)$$

$$= \frac{4}{Z} \Big[ \sum_{\mathbf{x}} \mathbf{f}(\mathbf{x}) \cdot \sum_{\mathbf{y} \sim \mathbf{x}} (\mathbf{f}(\mathbf{x}) - \mathbf{f}(\mathbf{y})) - \sum_{\mathbf{x}} \frac{\ell}{Z} |\mathbf{f}(\mathbf{x})|^2 \Big]$$

$$= 0.$$

where we use the fact that

$$\sum_{\mathbf{x}} \mathbf{f}(\mathbf{x}) \cdot \sum_{\mathbf{y} \sim \mathbf{x}} (\mathbf{f}(\mathbf{x}) - \mathbf{f}(\mathbf{y})) = \sum_{(\mathbf{x},\mathbf{y}) \in P} (\mathbf{f}(\mathbf{x}) - \mathbf{f}(\mathbf{y})) \cdot (\mathbf{f}(\mathbf{x}) - \mathbf{f}(\mathbf{y})) = \ell \qquad (34)$$

## J   Grokking on MNIST

To induce grokking on MNIST, we make two nonstandard decisions: (1) we reduce the size of the training set from 50k to 1k samples (by taking a random subset) and (2) we increase the scale of the weight initialization distribution (by multiplying the initial weights, sampled with Kaiming uniform initialization, by a constant $> 1$).

The choice of large initializations is justified by [31–33] which find large initializations overfit data easily but prone to poor generalization. Relevant to this, initialization scale is found to regulate "kernel" vs "rich" learning regimes in networks [34].

With these modifications to training set size and initialization scale, we train a depth-3 width-200 MLP with ReLU activations with the AdamW optimizer. We use MSE loss with one-hot targets, rather than cross-entropy. With this setup, we find that the network quickly fits the train set, and then much later in training validation accuracy improves, as shown in Figure 8a. This closely follows the stereotypical grokking learning, first observed in algorithmic datasets.

With this setup, we also compute a phase diagram over the model weight decay and the last layer learning rate. See Figure 8b. While in MLPs it is less clear what parts of the network to consider the "encoder" vs the "decoder", for our purposes here we consider the last layer to be the "decoder" and vary its learning rate relative to the rest of the network. The resulting phase diagram has some similarity to Figure 7. We observe a "confusion"phase in the bottom right (high learning rate and high weight decay), a "comprehension" phase bordering it, a "grokking" phase as one decreases weight decay and decoder learning rate, and a "memorization" phase at low weight decay and low learning rate. Instead of an accuracy threshold of 95%, we use a threshold of 60% here for validation accuracy for runs to count as comprehension or grokking. This phase diagram demonstrates that with sufficient regularization, we can again "de-grok" learning.

We also investigate the effect of training set size on time to generalization on MNIST. We find a result similar to what Power et al. [1] observed, namely that generalization time increases rapidly once one drops below a certain amount of training data. See Figure 20.

## K   Lottery Ticket Hypothesis Connection

In Figure 21, we show the projection of the learned embeddings after generalization to their first two principal components. Compared to the projection at initialization, structure clearly emerges in embedding space when the neural network is able to generalize ($> 99\%$ validation accuracy). What is intriguing is that the projection of the embeddings at initialization to the principal components of the embeddings at generalization seem to already contain much of that structure. In this sense, the structured representation necessary for generalization already existed (partially) at initialization. The training procedure essentially prunes other unnecessary dimensions and forms the required parallelograms for generalization. This is a nonstandard interpretation of the lottery ticket hypothesis

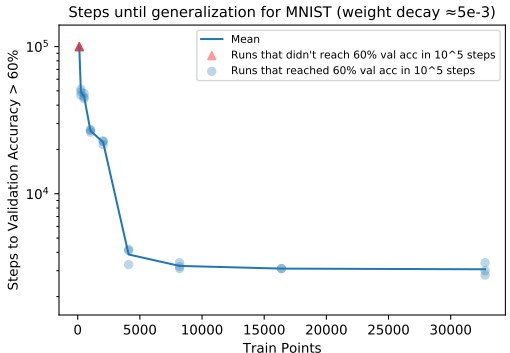

Figure 20: Time to generalize as a function of training set size, on MNIST.

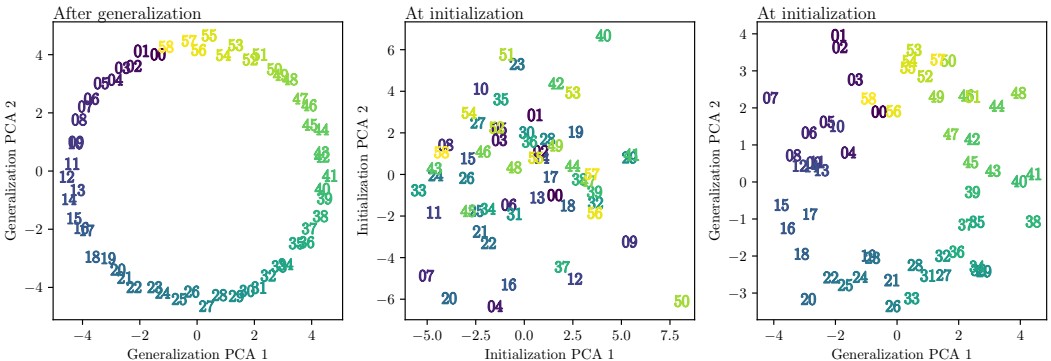

Figure 21: **(Left)** Input embeddings after generalization projected on their first 2 principal components.**(Center)** Input embeddings at initialization projected on their first 2 principal components. **(Right)** Input embeddings at initialization projected on the first 2 principal components of the embeddings after generalization at the end of training (same PCA as the left figure).

where the winning tickets are not weights or subnetworks but instead particular axes or linear combinations of the weights (the learned embeddings).

In Figure 22, we show the original training curves (dashed lines). In solid lines, we recompute accuracy with models which use embeddings that are projected onto the $n$ principal components of the embeddings at the end of training (and back). Clearly, the first few principal components contain enough information to reach 99% accuracy. The first few PCs explain the most variance by definition, however, we note that this is not necessarily the main reason for why they can generalize so well. In fact, embeddings reconstructed from the PCA at the end of training (solid lines) perform better than current highest variance axes (dotted line). This behavior is consistent across seeds.

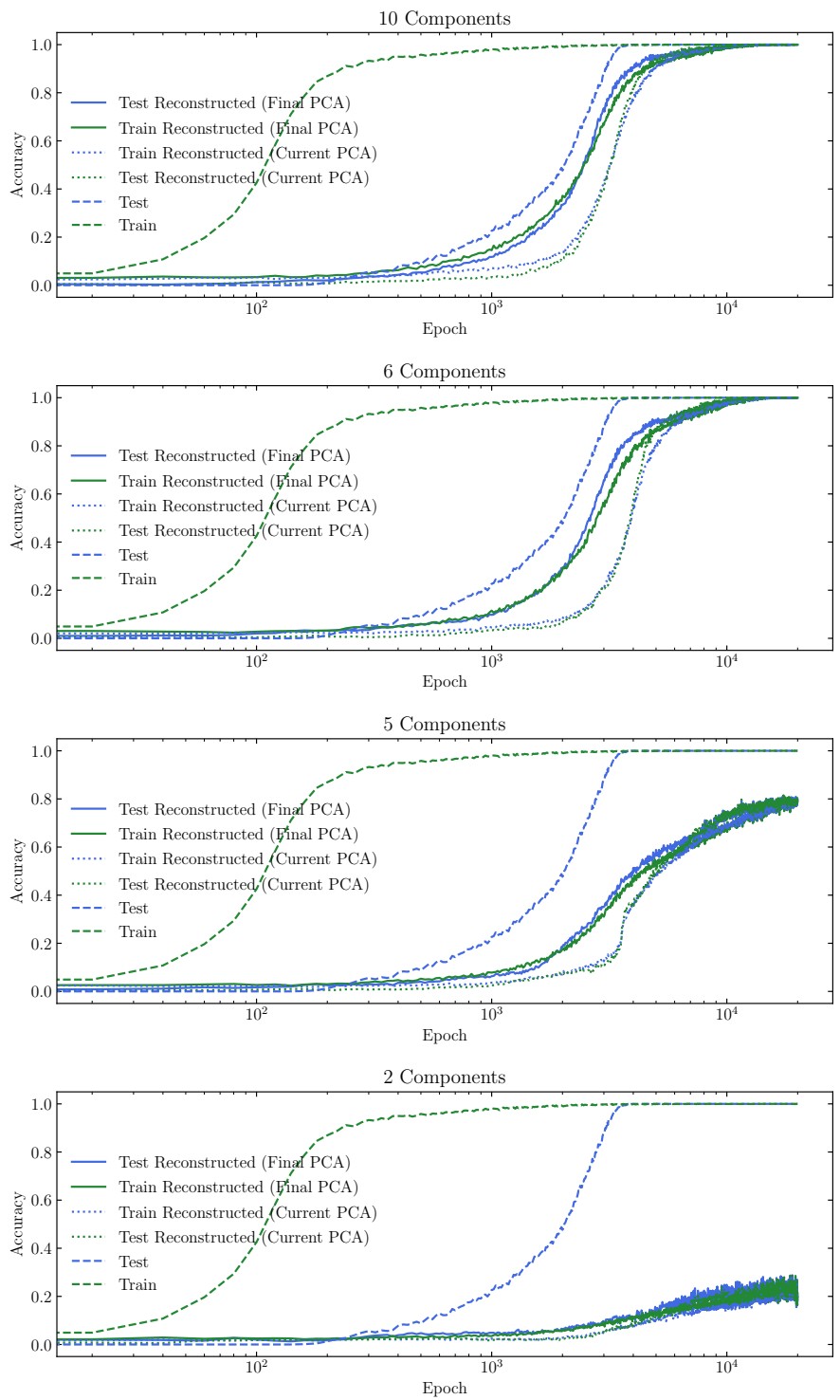

Figure 22: Train and test accuracy computed while using actual embeddings (dashed line) and embeddings projected onto and reconstructed from their first $n$ principal components (dotted lines) and, finally, using embeddings projected onto and reconstructed from the first $n$ PCs of the embeddings at the end of training (solid lines).

## L  Derivation of the effective loss

In this section, we will further motivate the use of our effective loss to study the dynamics of representation learning by deriving it from the gradient flow dynamics on the actual MSE loss in linear regression. The loss landscape of a neural network is in general nonlinear, but the linear case may shed some light on how the effective loss can be derived from actual loss. For a sample $\mathbf{r}$ (which is the sum of two embeddings $\mathbf{E}_i$ and $\mathbf{E}_j$), the prediction of the linear network is $D(\mathbf{r}) = \mathbf{A}\mathbf{r} + \mathbf{b}$. The loss function is ($\mathbf{y}$ is its corresponding label):

$$\ell = \underbrace{\frac{1}{2}|\mathbf{A}\mathbf{r} + \mathbf{b} - \mathbf{y}|^2}_{\ell_{\text{pred}}} + \underbrace{\frac{\gamma}{2}||\mathbf{A}||_F^2}_{\ell_{\text{reg}}}, \tag{35}$$

where the first and the second term are prediction error and regularization, respectively. Both the model $(\mathbf{A}, \mathbf{b})$ and the input $\mathbf{r}$ are updated via gradient flow, with learning rate $\eta_A$ and $\eta_x$, respectively:

$$\frac{d\mathbf{A}}{dt} = -\eta_A \frac{\partial \ell}{\partial \mathbf{A}}, \frac{d\mathbf{b}}{dt} = -\eta_A \frac{\partial \ell}{\partial \mathbf{b}}, \frac{d\mathbf{r}}{dt} = -\eta_x \frac{\partial \ell}{\partial \mathbf{r}}. \tag{36}$$

Inserting $\ell$ into the above equations, we obtain the gradient flow:

$$\begin{aligned}
\frac{d\mathbf{A}}{dt} &= -\eta_A \frac{\partial \ell}{\partial \mathbf{A}} = -\eta_A[\mathbf{A}(\mathbf{r}\mathbf{r}^T + \gamma) + (\mathbf{b} - \mathbf{y})\mathbf{r}^T], \\
\frac{d\mathbf{b}}{dt} &= -\eta_A \frac{\partial \ell}{\partial \mathbf{b}} = -\eta_A(\mathbf{A}\mathbf{r} + \mathbf{b} - \mathbf{y}) \\
\frac{d\mathbf{r}}{dt} &= -\eta_x \frac{\partial \ell}{\partial \mathbf{r}} = -\eta_x \mathbf{A}^T(\mathbf{A}\mathbf{r} + \mathbf{b} - \mathbf{y}).
\end{aligned} \tag{37}$$

For the $d\mathbf{b}/dt$ equation, after ignoring the $\mathbf{A}\mathbf{r}$ term and set the initial condition $\mathbf{b}(0) = \mathbf{0}$, we obtain analytically $\mathbf{b}(t) = (1 - e^{-2\eta_A t})\mathbf{y}$. Inserting this into the first and third equations, we have

$$\begin{aligned}
\frac{d\mathbf{A}}{dt} &= -\eta_A[\mathbf{A}(\mathbf{r}\mathbf{r}^T + \gamma) - e^{-2\eta_A t}\mathbf{y}\mathbf{r}^T], \\
\frac{d\mathbf{r}}{dt} &= \underbrace{-\eta_x \mathbf{A}^T \mathbf{A}\mathbf{r}}_{\text{internal interaction}} + \underbrace{\eta_x e^{-2\eta_A t}\mathbf{A}^T \mathbf{y}}_{\text{external force}}.
\end{aligned} \tag{38}$$

For the second equation on the evolution of $d\mathbf{r}/dt$, we can artificially decompose the right hand side into two terms, based on whether they depend on the label $\mathbf{y}$. In this way, we call the first term "internal interaction" since it does not depend on $\mathbf{y}$, while the second term "external force". Note this distinction seems a bit artificial from a mathematical perspective, but it can be conceptually helpful from a physics perspective. We will show below the internal interaction term is important for representations to form. Because we are interested in how two samples interact, we now consider another sample at $\mathbf{r}'$, and the evolution becomes

$$\begin{aligned}
\frac{d\mathbf{A}}{dt} &= -\eta_A[\mathbf{A}(\mathbf{r}\mathbf{r}^T + \mathbf{r}'\mathbf{r}'^T + 2\gamma) - e^{-2\eta_A t}\mathbf{y}(\mathbf{r} + \mathbf{r}')^T], \\
\frac{d\mathbf{r}}{dt} &= -\eta_x \mathbf{A}^T \mathbf{A}\mathbf{r} + \eta_x e^{-2\eta_A t}\mathbf{A}^T \mathbf{y}, \\
\frac{d\mathbf{r}'}{dt} &= -\eta_x \mathbf{A}^T \mathbf{A}\mathbf{r}' + \eta_x e^{-2\eta_A t}\mathbf{A}^T \mathbf{y}.
\end{aligned} \tag{39}$$

Subtracting $d\mathbf{r}/dt$ by $d\mathbf{r}'/dt$ and setting $\mathbf{r}' = -\mathbf{r}$, the above equations further simply to

$$\begin{aligned}
\frac{d\mathbf{A}}{dt} &= -2\eta_A \mathbf{A}(\mathbf{r}\mathbf{r}^T + \gamma), \\
\frac{d\mathbf{r}}{dt} &= -\eta_x \mathbf{A}^T \mathbf{A}\mathbf{r}.
\end{aligned} \tag{40}$$

The second equation implies that the pair of samples interact via a quadratic potential $U(\mathbf{r}) = \frac{1}{2}\mathbf{r}^T \mathbf{A}^T \mathbf{A}\mathbf{r}$, leading to a linear attractive force $f(r) \propto r$. We now consider the adiabatic limit where $\eta_A \to 0$.

**The adiabatic limit** Using the standard initialization (e.g., Xavier initialization) of neural networks, we have $\mathbf{A}_0^T \mathbf{A}_0 \approx \mathbf{I}$. As a result, the quadratic potential becomes $U(\mathbf{r}) = \frac{1}{2}\mathbf{r}^T\mathbf{r}$, which is time-independent because $\eta_A \to 0$. We are now in the position to analyze the addition problem. For two samples $\mathbf{x}^{(1)} = \mathbf{E}_i + \mathbf{E}_j$ and $\mathbf{x}^{(2)} = \mathbf{E}_m + \mathbf{E}_n$ with the same label $(i + j = m + n)$, they contribute to an interaction term

$$U(i, j, m, n) = \frac{1}{2}|\mathbf{E}_i + \mathbf{E}_j - \mathbf{E}_m - \mathbf{E}_n|_2^2. \tag{41}$$

Averaging over all possible quadruples in the training dataset $D$, the total energy of the system is

$$\ell_0 = \sum_{(i,j,m,n)\in P_0(D)} \frac{1}{2}|\mathbf{E}_i + \mathbf{E}_j - \mathbf{E}_m - \mathbf{E}_n|_2^2 / |P_0(D)|, \tag{42}$$

where $P_0(D) = \{(i, j, m, n)|i + j = m + n, (i, j) \in D, (m, n) \in D\}$. To make it scale-invariant, we define the normalized Hamiltonian Eq. (42) as

$$\ell_{\text{eff}} = \frac{\ell_0}{Z_0}, \quad Z_0 = \sum_i |\mathbf{E}_i|_2^2 \tag{43}$$

which is the effective loss we used in Section 3.2.