# OpenReview forum: "Towards Understanding Grokking: An Effective Theory of Representation Learning"
_NeurIPS.cc/2022/Conference — NeurIPS 2022 Accept_

### Official Review · Reviewer_Sd61 · 2022-07-08

**Rating:** 7
**Confidence:** 3
**Soundness:** 3 good
**Presentation:** 3 good
**Contribution:** 2 fair

**Summary:**

This work studies the Grokking phenomenon -- a phenomenon in which generalization occurs long after mastering the training set. This work poses three questions with regard to this phenomenon:
1. When the generalization happens? and the answer is: when the representations become well-structured.
2. How the size of the training set affects this phenomenon? and the answer is: the critical size of the dataset is the amount required for learning a 'good' representation.
3. What leads to a delay in generalization? and the answer is: different phases of learning between "memorization" (perfect training acc but with lack of generalization) and "comprehension" (perfect training and generalization acc)

From a physics perspective, this work build the answers outlined above and provide supporting experiments.

**Questions:**

1. The choice of Parallelogram-ratio for quantifying the quality of a representation is rather arbitrary. What is the idea behind using such a quantity?
2. Why don't you model the actual dynamics of GD on the training loss? Instead, you derive the dynamics of the 'effective' loss (defined in Eq. 5) which again appears arbitrary.
3. By defining the 'effective' loss, in essence the learning dynamics of the encoder is now decoupled from the decoder. I am right?
4. I have a hard time understanding lines 148-153. Would you explain why the third eigenvalue is the one that dominate the dynamics?
5. The analogy in lines 197-201; Don't you agree that it is the decoder that encourages the representations to have a structure rather than their "internal interactions". If the is not decoder, what forces the representations to change structure?

**Limitations:**

No negative societal impact is foreseen by this reviewer.
Authors talk about the limitations of the theory in line 160-165.

**Strengths And Weaknesses:**

**Strengths**:
- The problem is interesting and relevant: The Grokking phenomenon is a perplexing phenomenon which understanding it opens the doors to a better theory of generalization.
- The form of presentation is neat: I very much liked that the message is clear from the get go.
- The phase diagrams provide high-level intuitions: The same applies to the phase-diagrams that clearly distinguish between the 4 different phases.

** Weaknesses**:
- Some of the choices in the paper appears rather arbitrary and not fully justified (see questions below).
- At parts, it is difficult to follow certain arguments (at least for me): I am not from a physics back ground. For that it took me several wikipedia searches to grasp the meaning of some concepts in the paper. For example the term "effective" has a different meaning that the one that initially comes to mind.
- Very light on the related work: There has been a couple of papers that use tools from physics and provide phase diagrams for understanding the generalization of neural networks:
    + Generalisation error in learning with random features and the hidden manifold model by Gerace et al
    + Multi-scale Feature Learning Dynamics: Insights for Double Descent by Pezeshki et al
    + The Gaussian equivalence of generative models for learning with two-layer neural networks by Goldt et al
    + Statistical mechanics for neural networks with continuous-time dynamics by Kuhn et al
- Very limited experimental results.

---

> ### Author Response · Authors · 2022-08-02
> **Response to Sd61 (Part I)**
>
> Dear Reviewer Sd61 -- Thank you very much for your review, especially for the references to the related papers! Below we address your concerns and questions. Please let us know if something remains unclear.
>
> ### Weaknesses
> _"At parts, it is difficult to follow certain arguments. For example, the term "effective" has a different meaning that the one that initially comes to mind."_
> * We apologize for the confusion. The concept of an effective theory in physics is similar to model reduction in computational methods, aiming to describe complex phenomena with a simpler theory which is tractable to analyze but still captures the basic behavior being studied. In our revised manuscript, we have clarified the definition at the beginning of Section 3.
>
> _"Cite related works"_
> * Thanks for pointing to these nice papers!. We have added them in the related works.
>
> _"Limited experimental results"_
> * We have added more experiments to strengthen our theory/method. In particular, we show that both our effective theory and phase diagrams can generalize to image classifications (run on the MNIST datasets). Please consider the new Appendices G, H, and I, marked blue in the revised paper!
>
> ### Questions
> _"1. The choice of Parallelogram-ratio for quantifying the quality of a representation is rather arbitrary. What is the idea behind using such a quantity?"_
> * Thanks for pointing that out, we should be more clear on that! We have made the motivations more clear in the paper.
> * The intuition is as follows:
>     * (i) the numbers are embedded into a vector space
>     * (ii) if $E_1 + E_4 = E_2 + E_3$ (with $E_X$ being the embedding of number $X$), the learning process correctly placed those embeddings relative to each other.
>     * (iii) Parallelograms are explicit causes of generalization. For example, Suppose the pair (1, 4) appears in the training set, and the pair (2, 3) appears in the validation set. If $(E_1, E_2, E_3, E_4)$ forms a parallelogram, i.e., $E_1 + E_4 = E_2 + E_3$,  then the training sample $(E_1, E_4)$ can immediately generalize to the validation sample $(E_2, E_3)$. If all possible parallelograms are set up correctly,
> i.e. $E_X + E_Y = E_Z + E_W$ for all $X,Y,Z,W$ where $X+Y = Z+W$, then the embeddings will lie on one line and the scalar addition of the inputs becomes equivalent to the vector addition in the embedding space and we have perfect generalization. This is why the fraction of parallelograms learned is a good indicator of the generalization capability of the network.
>
> _"2. Why don't you model the actual dynamics of GD on the training loss? Instead, you derive the dynamics of the 'effective' loss (defined in Eq. 5) which again appears arbitrary & 3. By defining the 'effective' loss, in essence the learning dynamics of the encoder is now decoupled from the decoder. I am right?"_
> * Yes -- in reality, gradients for the embeddings are backpropagated through the decoder MLP. However, it turns out that we can model the learning dynamics of the embeddings as if they didn’t depend on the details of the decoder, but instead just on their relative positions. With this simplification, learning dynamics become much easier to theoretically analyze. Despite this simplification, we find that the dynamics under the effective loss capture some important properties of the true model, notably the dependence on the train data fraction. Lastly, we’ll note here that one can derive our “effective loss” from the true loss in a certain limit, with some assumptions, as demonstrated in Appendix K of our revised manuscript.

---

> > ### Author Response · Authors · 2022-08-02
> > **Response to Sd61 (part II)**
> >
> > _"3. I have a hard time understanding lines 148-153. Would you explain why the third eigenvalue is the one that dominate the dynamics?"_
> > * Of course!
> > We say a representation is a solution if it incurs zero effective loss.
> > As already discussed above, a line in embedding space, $E_k = a + k b$, does that.
> > * Now we look at the solution $R(t)$ to the differential equation (9). We know that $H$ has at least two zero eigenvalues. All contributions to $R(t)$ with non-zero $\lambda_i$ will go to 0 with increasing time, while the first two contributions remain constant. The two remaining eigenvectors will define the line $E_k$. As the other eigenvalues "interfere" with the line representation, we have to wait for them to decay with time. And since $\lambda_3$ is the smallest one, that decay will take the longest. Thus, the size of $\lambda_3$ determines the time for generalization. If $\lambda_3$ is also 0, we will never get to a line representation and thus not achieve perfect generalization. That defines the critical training set size. See Figure 17(a) for how $\lambda_3$ depends on the training set size.
> >
> >
> > _"4. The analogy in lines 197-201; Don't you agree that it is the decoder that encourages the representations to have a structure rather than their "internal interactions". If the is not decoder, what forces the representations to change structure?"_
> > * The effective loss only contains so-called two-body interaction terms i.e. terms that depend on the relative positions of the particles (embeddings). This is what we meant by internal interactions. You are correct that the decoder determines the dynamics under real training. However, we find that our effective theory, which is independent of the decoder, nevertheless captures the core behavior of the learning dynamics. We updated the language of the draft for clarity.

---

### Official Review · Reviewer_tEaJ · 2022-07-12

**Rating:** 7
**Confidence:** 4
**Soundness:** 3 good
**Presentation:** 3 good
**Contribution:** 2 fair

**Summary:**

This paper studies the grokking phenomenon under a physics-inspired "effective theory", which studies the phenomenon under toy settings. The grokking phenomenon refers to when generalization happens much later during training, long after training accuracy have reached 100%. The theory proposes the following explanations to grokking.
1) generalization is correlated to when the models learns a sensible representation of the inputs.
2) the critical training size (amount of data needed for generalization) relates to the amount of data needed to determine a good representation.
3) grokking represents a particular region in phase space of hyperparameters, and can be removed by using better hyperparameters.

**Questions:**

- Is the analysis on "time towards the linear structure" independent of dataset size? Or can it shed light on the question of data size vs grokking / comprehension?

- What type of task characteristic makes a model most susceptible to grokking? If the mismatch between decoder learning speed and representation learning speed is key to grokking, would you expect that the four phases in the phase diagram to exist in general (e.g. for natural vision or nlp tasks)? If it's not general, what is it about the specific tasks and models studied in this work that leads to these different regions?

**Limitations:**

The authors address the limitations of the toy model. Though it is fine to use toy models to study these type of questions, I think that there is nonetheless insufficient work to show whether the insights from the toy model can actually generalize to other settings.

**Strengths And Weaknesses:**

### Strength:

- The paper proposes an original view into the grokking phenomenon, which has been an interesting and unexplained phenomenon in the community. The phase diagram view into the landscape of training regimens illustrates interesting relationships between memorization, generalization, and grokking. One of the interesting questions about grokking is whether there is some magical inductive bias of gradient descent that leads to generalization after an abnormally long training period. The result of this paper would suggest that grokking is more of a pathological phenomenon that is the result of improper hyperparameter tuning. Though as I'll mention in the weakness section, I think that the support for this argument is not sufficient if it is indeed the conclusion of the authors.

- The paper proposes an explanation into the grokking phenomenon in the form of competition between representation learning and decoder overfitting. This seems plausible and is in line with a number of other works suggesting that later layers drive overfitting.

- Although the theory is based on a very simplified toy model, it provides an illuminating view into why a minimum number of training examples may be required for generalization. The predicted phase transition in Figure 4 is interesting to see.

### Weaknesses:

- One of the intriguing observations from Power et al [1] is that grokking can be seen both in unregularized models with sufficient data, or in regularized models with a small amount of training data. I think a sufficient explanation of the phenomenon needs to provide a justification for both behavior. However, the discussion around the effect of data size seems lacking. As far as I can tell, neither the theoretical nor the empirical analyses shed light on why more data speeds up generalization while less data can lead to grokking.

- Most of the results use an extremely toy setting. Only one set of experiments (modulo addition with varying learning rate and weight decay) is performed on the original grokking task from [1], while [1] contains a number of tasks and hyperparameter settings. Crucially the effect of data size is not studied, which was an important observation from [1]. It is unclear how much this theory really explains the original phenomenon, particularly given that the behavior of the phase diagram is qualitatively different between the setting in [1] and the toy model, namely that instead of models going from memorization to comprehension to grokking, in the real task setting it is going from memorization to grokking to comprehension. There is insufficient explanation to bridge this gap.

---

> ### Author Response · Authors · 2022-08-02
> **Response to tEaJ**
>
> Dear Reviewer tEaJ -- Thank you for reviewing our paper and for your detailed feedback!
> Below, we address your questions. Please let us know if anything remains unclear.
>
> ### Weaknesses
>
> Q: _"One of the intriguing observations from Power et al [1] is that grokking can be seen both in unregularized models with sufficient data, or in regularized models with a small amount of training data. I think a sufficient explanation of the phenomenon needs to provide a justification for both behaviors. However, the discussion around the effect of data size seems lacking. As far as I can tell, neither the theoretical nor the empirical analyses shed light on why more data speeds up generalization while less data can lead to grokking."_
> * A: Our effective theory does indeed capture dependence of grokking time on data set size, although this was not adequately explained in our original manuscript. We have added new discussion and results on this in Appendix G.
> * The basic idea is this: We argue that the learning speed in our setting is governed by $\lambda_3$, the third-smallest eigenvalue of $A^T A = H$. $A$ is a matrix representing the parallelogram equations, see Equation 7 in Section 3.2. The larger $\lambda_3$, the faster the training. In the plot in Appendix G, we show the size of $\lambda_3$ depends on the train data fraction. In Figure 4b one can find the empirical results for the toy model.
> * Additionally, we show empirically how the size of the training dataset influences grokking/delayed generalization in image classification, see Appendix I, Figure 20.
>
> Q: _"The behavior of the phase diagram is qualitatively different between the setting in [1] and the toy model, namely that instead of models going from memorization to comprehension to grokking, in the real task setting it is going from memorization to grokking to comprehension. There is insufficient explanation to bridge this gap."_
> * A: Thanks for pointing that out, we should have been clearer here! The phase diagram's main point is to show that grokking is pathological, a result of improperly tuned hyperparameters.  Going from X to Y implies choosing a path on the plane, and depending on the choice the results are different. In that sense, one can almost always find a way in which everything is adjacent to everything else, as long as they have a common border. Note also that the phase diagrams only show a part of the plane!
> * We admit that the phase diagrams in Figure 5 and 6 are qualitatively different. This could be due to several factors. One possibility is that it is due to differences in embedding dimensions (1D and 256D, respectively). While 1D prefers linear representation, 256D prefers the ring representation.
>
> ### Questions
> Q: _"Is the analysis on "time towards the linear structure" independent of dataset size? Or can it shed light on the question of data size vs grokking / comprehension?"_
> * A: It can indeed shed light on the dependence of grokking on data size. The dynamics of effective theory is dependent on the training data size, since the effective loss is the sum of four-body terms within the training set. As we discuss in Appendix G, the time of grokking scales as $1/\lambda_3$, and we find that $\lambda_3$ is (on average) zero below some train set fraction and then an increasing function of of the train set fraction (Figure 17a), so grokking time decreases as data size grows larger.
>
> Q: _"What type of task characteristic makes a model most susceptible to grokking? If the mismatch between decoder learning speed and representation learning speed is key to grokking, would you expect that the four phases in the phase diagram to exist in general (e.g. for natural vision or nlp tasks)? If it's not general, what is it about the specific tasks and models studied in this work that leads to these different regions?"_
> * A: We expect that four phases of learning to exist in general, but the ease of obtaining them in practice depends on specific tasks. We think grokking is easy to get on datasets in which initial representation is very far from a good, final representation. From comparing (a) arithmetic datasets and (b) image classification: (a) the embeddings of numbers are initialized as random vectors, thus far from the desired, structured representations. (b) Images, although pixelization may destroy or obfuscate some semantic information, other information (e.g. topology of  the image manifold) is still preserved, making it faster to learn a good representation. In fact, we are able to observe grokking on MNIST classification, if we manually construct a suboptimal initial representation by using a large initialization scale. The dependence on training set size and the phase diagrams are consistent with theory developed in the paper, see Appendix I.

---

### Official Review · Reviewer_k44Q · 2022-07-15

**Rating:** 6
**Confidence:** 4
**Soundness:** 3 good
**Presentation:** 3 good
**Contribution:** 3 good

**Summary:**

The main contribution of this paper is a study of how different loss minimization and generalization outcomes are realized: confusion (no loss minimization), memorization (loss minimization without generalization), fast generalization, and grokking (generalization, but significantly slower than the loss is minimized). There is a secondary result on the minimal training set size required to induce a representation in which different classes are linearly separable. All tasks studied in the paper are related to arithmetic operations like addition, with an appendix explaining that the setup studied in the paper can be used for all Abelian groups. The paper contains a thorough exploration of a toy system, which is used to analyze and illustrate the above-mentioned outcomes. Some results are included on a larger task with larger networks.

**Questions:**

- Can you be a bit more explicit about the loss in the beginning? It would help reading the paper.
- If I read the paper correctly, you are saying that grokking happens when different parts of the network absorb information at different rates; in particular, when the last part (the decoder) is much faster than the preceding part. Is that a reasonable formulation? And if yes, is it possible to quantify that statement?
- What constitutes a part, for the purposes of the above question? (see also discussion of A1 in the ‘strengths and weaknesses’ section)
- How generally does your analysis apply? I can see from the appendices that it should generalize to other Abelian groups, but can you comment on different applications like image classification? The answers you give - linear representation, relative learning speeds of the decoder and upstream subnetwork - are not specific to the type of problem you study, so it would be very interesting (and make the paper much stronger) if you could also include results on other types of task.
- How strongly does the result for the critical training set size to obtain a linear representation depend on the simple effective loss you are using? Would it be possible to perform a similar analysis for image classification (or language modelling, or any other application you fancy)?
- Regarding the p=53 modulo addition experiment: is there a linear subspace in which you find the pizza wheel representation, like for the toy task? I understand it’s hard to visually inspect the 256D embeddings, but it might be interesting to see how many ‘effective dimensions’ there are - e.g. how many PCA components have significant explanatory power. It would be reasonable to expect a low-dimensional effective embedding, given how neatly t-SNE puts all the numbers in a circle. What happens if you take the two most explanatory principal components and make the same decoder plot there that you have for the toy task?
- How do your findings relate to the lottery ticket hypothesis? That also addresses generalization in unexpected situations - in overparameterized networks as opposed to late in training. Would having more highly overparameterized representation networks be another potential way to avoid grokking, given the lottery ticket hypothesis that bigger networks have higher chances of containing subnetworks with good initializations, that would therefore learn quickly?
- In the particle system analogy: What are the ‘internal particle interactions’? If they arise from the gradients computed from the loss, then they really originate ‘externally’ too, don’t they?


**Limitations:**

The main limitation comes from the size of the systems studied, as mentioned above. In principle the analysis could apply quite widely, but it is difficult to assess that from the experiments in the paper.

**Strengths And Weaknesses:**

- Strength: the exploration of the toy system is very thorough and offers an interesting ‘peek inside the black box’. It suggests very plausible mechanisms for how the learning process plays out.
- Weakness: the paper is quite limited in terms of bigger systems, which makes it difficult to assess how general the results are.
- Weakness: The current formulation of A1 (the answer to question Q1, when does generalization happen), while true, seems both a bit trivial and a bit vague. You say generalization happens ‘with a good representation’, or a representation whose ‘structure is appropriate for the task’. This answer leaves open the questions which representation needs to be good (is it the penultimate layer of the network? the input to the decoder? but where does the decoder start and the representation network end?) and when a representation is ‘good’ or a structure ‘appropriate’. Presuming that those questions are answered, it comes down to the statement that the mapping from the representation to the final required output is easy to learn. For the toy task you analyze you do go into the structure of the representation quite extensively, but since that analysis is limited to the toy setup, it is not obvious that the answers you find there generalize to harder / bigger tasks, deeper networks, and higher-dimensional representations. NB: A2 is only as meaningful as A1 is clear, which makes this issue more relevant.

---

> ### Author Response · Authors · 2022-08-02
> **Response to k44Q (Part I)**
>
> Dear Reviewer k44Q -- Thank you for your detailed review and constructive feedback! Here is our response to your questions and concerns:
>
> **Summary**
>
> **Weakness:**_"[...] There is a secondary result on the minimal training set size required to induce a representation in which different classes are linearly separable."_
> * Just to clarify here: In our discussion of the toy model (Section 3), when we refer to a “linear structure”, we mean that the embeddings are arranged along a straight line (i.e. $E_k = a + kb$ where a and b are vectors and k is an integer.), rather than that they are linearly separable. In the toy model for addition, this arrangement of the embeddings leads directly to generalization -- see Section 3.1.
>
> **Weakness:** _"the paper is quite limited in terms of bigger systems, which makes it difficult to assess how general the results are."_
> * Our primary goal with this paper was to provide theoretical insight into the empirical results of Power et al., who first observed grokking in a relatively narrow domain: relatively small neural networks being trained on small algorithmic datasets. Accordingly, we have focused on this relatively small-scale setting.
> * However, we are also interested in whether our analysis of grokking extends to more general tasks/architectures. To address your concern, we have added more experiments which show that both our effective theory and phase diagrams can generalize to image classification. We include the new results in Appendix H and I, including the first demonstration, to our knowledge, of grokking on an image classification task. While our image classification setting is still relatively small-scale (MNIST), we believe these new results will be of significant theoretical interest, since they demonstrate that grokking is a more general phenomenon in deep learning than was previously known.
>
> **Weakness:** _"The current formulation of A1 (the answer to question Q1, when does generalization happen), while true, seems both a bit trivial and a bit vague. You say generalization happens ‘with a good representation’, or a representation whose ‘structure is appropriate for the task’. This answer leaves open the questions which representation needs to be good (is it the penultimate layer of the network? the input to the decoder? but where does the decoder start and the representation network end?) and when a representation is ‘good’ or a structure ‘appropriate’."_
> * Representations, in any layer of a network, are good insofar as they allow the rest of the network to easily compute, in a manner that generalizes, the desired output from that representation. Exactly what constitutes a good representation thus depends on the network and on the task. Empirically, we observe that ordered ring structures in the embeddings of transformers are correlated with generalization on tasks like modular arithmetic, the setting of Power et al. Given some network, figuring out what part to consider the encoder vs the decoder is an empirical question -- almost a matter of interpretability, answered by carefully studying the internal workings of trained models. In transformers, since we observed intricate structure emerge in the embeddings (Figure 1), we considered the model embedding to be the encoder and the transformer “decoder” to be the decoder. Once a decision has been made, one can then study how various hyperparameters affect the formation of structure within the network.
> * In our toy model for addition, where addition between embeddings is part of the architecture, the linear structure in the embeddings is a good representation. This is the advantage of the toy model -- one can directly see that the embeddings are the representations that matter, and that a particular structure (a linear structure) is good, since it guarantees generalization.

---

> > ### Author Response · Authors · 2022-08-02
> > **Response to k44Q (Part II)**
> >
> > **Questions**
> >
> > _"Can you be a bit more explicit about the loss in the beginning? It would help reading the paper."_
> > * We have updated the explanation of the losses we use in the revised version of the paper.
> > * To clarify, we would like to distinguish two kinds of losses used in our work: (i) actual training loss (mean-squared error or cross entropy) to obtain empirical results; (ii) effective loss, as a simplification of the actual training loss.  We use the term "effective loss" to invoke the idea of an “effective theory” in physics -- a theory which describes a system’s behavior without fully modeling the true causes of that behavior.
> > * One can think about our effective loss as a postulated (approximate) implicit loss induced by the training procedure (the architecture, the actual loss being minimized, the optimizer used, etc.). Its purpose is to model the training dynamics of representations in a way that can be more directly analyzed. In particular, the mechanism described in Section 3 is that the neural network will try to simply learn parallelograms in embedding space. This simple view seems to be able to predict numerous phenomena: the criticality of training set size for generalization, the structure of the learned representations in embeddings space, and, to a lesser extent, the dynamics of the embeddings throughout training.
> >
> > _"If I read the paper correctly, you are saying that grokking happens when different parts of the network absorb information at different rates; in particular, when the last part (the decoder) is much faster than the preceding part. Is that a reasonable formulation? And if yes, is it possible to quantify that statement?"_
> > * This is a reasonable formulation. Looking at phase diagrams in Figure 5 and 6, grokking occurs for a particular combination of decoder learning rate and weight decay, which we consider two of the major contributors to “learning speed”. We believe the phase diagrams shed some light on the roles of these hyperparameters. We cannot quantify the statement beyond our empirical results yet. However, the main conclusion should be that grokking is pathological and can be remedied by changing hyperparameters.
> >
> > _"What constitutes a part, for the purposes of the above question? (see also discussion of A1 in the ‘strengths and weaknesses’ section)"_
> > * We have defined the encoder to be the learned embedding while the decoder is the rest of the neural network, for most of our experiments. As discussed earlier, this depends on the task, but it is well-justified theoretically for the toy model and well-justified empirically for transformers being trained on algorithmic tasks.
> >
> > _"How generally does your analysis apply? [...] The answers you give - linear representation, relative learning speeds of the decoder and upstream subnetwork - are not specific to the type of problem you study, so it would be very interesting (and make the paper much stronger) if you could also include results on other types of task."_
> > * As the question noted, our analysis applies to arithmetic datasets in which the Grokking phenomenon was first observed. One reason for the focus on arithmetic datasets, and specifically on the toy setting, is precisely because one can predict the proper representations necessary for generalization -- linear for addition and circular for modular addition. In the algorithmic datasets, we know exactly what to expect as a perfect representation, that is the representation that makes the task "as easy as possible" for the decoder.
> > In the addition setting, for instance, the best way to encode numbers is along a line, such that addition in the representation space can simply be achieved exactly and generally by adding the embedding vectors. To summarize, we started with problems that humans can solve, since in these special cases we know what to expect.
> > * For tasks like image classification, defining a perfect representation is a much harder task.
> > * However, your question prompted us to try to apply parts of our analysis to image classification, with good results. We include the effective theory analysis in Appendix H, and phase diagram analysis in Appendix I. To summarize our findings here: (i) The effective theory predicts the training dynamics similar to the neural collapse phenomenon [1]. (ii) The effective theory gives rise to a novel representation learning method (iii) However, the effective theory is unable to determine the exact critical training set size, see the question below. (iv) MNIST classification manifests all four learning phases (including grokking), and its phase diagram is very similar to that of algorithmic datasets. The resemblance seems to imply the universality of our phase diagram analysis, though additional studies on other types of problems are still needed to confirm universality

---

> > > ### Author Response · Authors · 2022-08-02
> > > **Response to k44Q (Part III)**
> > >
> > > _"How strongly does the result for the critical training set size to obtain a linear representation depend on the simple effective loss you are using? Would it be possible to perform a similar analysis for image classification (or language modeling, or any other application you fancy)?"_
> > > * The critical training set size does not depend on the loss, per se. The theoretical model offers an explanation for the existence of a critical set in the toy setting i.e. a phase transition in the probability of obtaining a linear structure (which would lead to generalization). The argument here is that different operations  (for various algorithmic datasets) will have similar behavior based on a generalized version of parallelograms and "linear structure." This is something we can already visualize in the modular addition case in Figure 1.
> > > Determining the critical training set size requires the knowledge of ground truth representation, which is not well-defined in image or language tasks. It may require more extensive domain knowledge to do the critical size analysis, which we would like to investigate in the future.
> > >
> > > _"Regarding the p=53 modulo addition experiment: is there a linear subspace in which you find the pizza wheel representation, like for the toy task? I understand it’s hard to visually inspect the 256D embeddings, but it might be interesting to see how many ‘effective dimensions’ there are - e.g. how many PCA components have significant explanatory power. It would be reasonable to expect a low-dimensional effective embedding, given how neatly t-SNE puts all the numbers in a circle. What happens if you take the two most explanatory principal components and make the same decoder plot there that you have for the toy task?"_
> > > * Figure 1 shows exactly that! There we plot the first two principal components from a PCA on the embeddings. (N.b., the fact that modular addition can be represented by a circle like this is why we chose this example.) We also added plots showing that a small number of principal components of the input embeddings is enough to reproduce perfect generalization.
> > >
> > > _"How do your findings relate to the lottery ticket hypothesis? That also addresses generalization in unexpected situations - in overparameterized networks as opposed to late in training. Would having more highly overparameterized representation networks be another potential way to avoid grokking, regiven the lottery ticket hypothesis that bigger networks have higher chances of containing subnetworks with good initializations, that would therefore learn quickly?"_
> > > * This is an interesting thought! We’ve added some discussion of the LTH to Appendix J. While the connection to LTH is not perfect, we observe that along the right components, the embeddings are already roughly structured at initialization. So perhaps one could view generalization (grokking) as a process where superfluous structure is pruned out by SGD, leaving behind simple subnetworks which generalize. The connection between LTH and grokking is definitely a promising direction for future investigation.
> > >
> > > _"In the particle system analogy: What are the ‘internal particle interactions’? If they arise from the gradients computed from the loss, then they really originate ‘externally’ too, don’t they?"_
> > > * The effective loss only contains so-called two-body interaction terms i.e. terms that depend on the relative positions of the particles (embeddings). This is what we meant by internal interactions. We updated the draft to reflect this.
> > >
> > > [1] Papyan, Vardan, X. Y. Han, and David L. Donoho. "Prevalence of neural collapse during the terminal phase of deep learning training." Proceedings of the National Academy of Sciences 117.40 (2020): 24652-24663.

---

> > > > ### Comment · Reviewer_k44Q · 2022-08-09
> > > > **Increasing score to 6**
> > > >
> > > > Thank you for your extensive rebuttal. You have addressed a number of my questions and concerns, and I am increasing my score to a 6: weak accept. I appreciate the new appendices, they add extra depth and understanding to the paper. The one remaining concern is scale: results on large-scale systems would still add considerably to the paper’s contribution. However, as it stands I would argue in favor of accepting the paper, because it reveals interesting dynamics in the learning process that are likely to also be present at larger scales.
> > > >
> > > > Regardless of whether the paper gets accepted eventually, I would encourage you to work on showing that your findings here are relevant to training large-scale systems. I believe they might very well be, even if that is outside the scope of your current submission.
> > > >
> > > > You say, rightly, that a perfect representation is harder to define for more complex data. That is indeed the reason to use toy/small-scale datasets, but in order for the findings on small-scale data to be relevant to large-scale settings, and hence to the research community as a whole, it is very important to show that the conclusions generalize to situations that are harder to inspect.
> > > >
> > > > Apologies for my misreading of figure 1 - I thought the circular arrangement was found through t-SNE instead of PCA. Thanks for the clarification.
> > > >
> > > > I have another question or remark about the MNIST results, just for your consideration: The relative ordering of the phases is different than in the arithmetic task: memorization and confusion do not border each other, while memorization and grokking do (they also do in some places in the arithmetic task, but not to the same extent as in the MNIST case). The explanation in terms of absorbing information at different rates sounds like it would suggest a particular phase diagram, which one might expect to be consistent between different tasks, since the explanation does not refer to the task. It would be interesting to understand this in more detail.

---

### Author Response · Authors · 2022-08-02
**Summary of Updates**

Firstly, we want to thank all reviewers for their high-quality reviews and constructive feedback. This first round of reviews prompted us to make a number of improvements to our manuscript. The revised version not only contains improvements in clarity, but also adds substantial new analysis and results that we believe make our work materially stronger and also much more general. In the updated manuscript, major revisions, as well as passages relevant to our answers, are highlighted in blue. Here is a summary of these revisions/additions:

**1. Better justification and analysis of our “effective theory”**
* We add additional discussion and results showing how our effective theory predicts the dependence of grokking on training set size. See Appendix G for more details.
* We clarify the language in Section 3 for what we mean by “effective theory” and “linear structure”.
* In response to reviewer Sd61, we hope to motivate the effective loss a little better by showing how it can be derived in the linear regression setting from actual gradient flow dynamics. See Appendix K.

**2. Extension of our analysis beyond algorithmic datasets**
A primary concern of reviewer k44Q was whether our analysis was too limited in scope. They mentioned image classification as an application where we could extend our analysis, and suggested that doing so would make our paper much stronger. This was a fruitful suggestion, which led to the following:
* We develop an effective loss function for image classification problems. This effective theory gives rise to a self-supervised learning method free of global collapse. It also aligns with the phenomenon of neural collapse. We visualize how representations evolve under this effective loss on the MNIST dataset. See Appendix H.
* We demonstrate that grokking (significantly delayed generalization) can occur in networks trained on MNIST. To our knowledge, this is the first time that grokking has been observed beyond the algorithmic datasets studied in Power et al. We include experimental results on MNIST in Appendix I. These include a phase diagram as well as a study of how time to generalization depends on the amount of training data. We find that grokking in this setting can again be remedied using the same prescription we develop for algorithmic datasets -- with proper tuning of weight decay and learning rates.

We have also made some miscellaneous other edits. For instance, in order to both respond to reviewer concerns and also stay within the page limit, we have removed some parts of “Related Work”. We want to add these back for the “camera ready” version.

---

### Meta-Review · Area_Chair_dhEp · 2022-08-21

**Recommendation:** Accept
**Confidence:** Certain

**Metareview:**

There was a consensus among reviewers that this paper should be accepted. In particular, reviewers felt, that the contribution of studying the rate at which different parts of a neural net absorb information, and the effects it has on learning and generalization is a worthwhile one that would enrich the literature and that the paper is a solid contribution to the NeurIPS literature.

**Award:**

No

---

### Decision · Program_Chairs · 2022-09-14

Accept